# Application of Chemistry-climate model SOCOL-AERv2-BEv1 to cosmogenic beryllium isotopes: Description and validation for polar regions

Kseniia Golubenko[1], Eugene Rozanov[2,3,4], Gennady Kovaltsov[5], Ari-Pekka Leppänen[6],
Timofei Sukhodolov [3,4], and Ilya Usoskin[1,7]

[1]Space climate research unit, University of Oulu, Oulu, 90570, Finland
[2]Physikalisch-Meteorologisches Observatorium Davos and World Radiation Center, Davos Dorf, 7260, Switzerland
[3]Institute for Atmospheric and Climate Science, ETH Zurich, Zurich, 8092, Switzerland
[4]St. Petersburg State University, St. Petersburg, 198504, Russia
[5]Ioffe Physical-Technical Institute, St. Petersburg, 194021, Russia
[6]Radiation and Nuclear Safety Authority – STUK, Rovaniemi, 96400, Finland
[7]Sodankylä Geophysical Observatory, Sodankylä, Oulu, 99600, Finland

**Correspondence:** Kseniia Golubenko (kseniia.golubenko@oulu.fi)

**Abstract.** Short-living cosmogenic isotope $^7$Be, produced by cosmic rays in the atmosphere is often used as a tracer for atmospheric dynamics where precise and high-resolution measurements cover the recent decades. Long-living isotope $^{10}$Be, measured in polar ice cores with an annual resolution, makes a proxy for long-term cosmic-ray variability, whose signal can, however, be distorted by atmospheric transport and deposition, that need to be properly modelled to be accounted for. While transport of $^7$Be can be modelled with high accuracy using the known meteorological (wind) fields, atmospheric transport of $^{10}$Be was typically modelled using case-study specific simulations or simplified box models based on parametrizations. Thus, there is a need for a realistic model able to simulate atmospheric transport and deposition of beryllium with a focus on polar regions and (inter)annual time scales and potentially able to operate in a self-consistent mode, without the prescribed meteorology. Since measurements of $^{10}$Be are extremely laborious, and hence scarce, it is difficult to compare model results directly with measurement data. On the other hand, the two beryllium isotopes are believed to have similar transport/deposition properties being different only in the production and the lifetime, thus the results of $^7$Be transport can be generally applied to $^{10}$Be. Here we present a new model to trace isotopes of $^7$Be and $^{10}$Be in the atmosphere based on the chemistry-climate model SOCOL (SOlar Climate Ozone Links) v3, which has been improved by including modules for the production, deposition, and transport of $^7$Be and $^{10}$Be. Production of the isotopes was modelled for both galactic and solar cosmic rays, by applying the CRAC (Cosmic-Ray induced Atmospheric Cascade) model. Transport of $^7$Be was modelled without additional gravitational settling due to the sub-micron size of the background aerosol particles. An interactive deposition scheme was applied including both wet and dry depositions. Modelling was performed using a full nudging to the meteorological fields, for the period of $2002-2008$ with a spin-up period of $1996-2001$. The modelled concentrations of $^7$Be in near-ground air were compared with the measured ones at a weekly cadence, in four nearly antipodal high-latitude locations, two in Northern (Finland and Canada) and two in Southern (Chile and Kerguelen Island) hemispheres. The model results agree with the measurements in

the absolute level within error bars, implying that the production, decay, and lateral deposition are correctly reproduced. The model also correctly reproduces the temporal variability of $^7$Be concentrations on the annual and sub-annual scales, including the presence/absence of the annual cycle in the Northern/Southern hemispheres, respectively. We also modelled the production and transport of $^7$Be for a major solar energetic-particle event (SPE) of 20-Jan-2005, which appears insufficient to produce a measurable signal but may serve as a reference event for historically known extreme SPEs. Thus, a new full 3D time-dependent model, based on the SOCOL v3.0, of $^7$Be and $^{10}$Be atmospheric production, transport and deposition have been developed. Comparison with the real data of $^7$Be concentration in the near-ground air validates the model and its accuracy.

## 1   Introduction

One of the most important outer-space factors affecting Earth is related to cosmic rays (highly energetic nuclei of extra-terrestrial origin), which cause nucleonic-electromagnetic-muon cascade in the terrestrial atmosphere (Dorman, 2004). These cosmic-ray induced atmospheric cascades form the main source of ionization in the troposphere and stratosphere (e.g., Mironova et al., 2015). Nuclear interactions between the nucleonic component of the cascade and nuclei of atmospheric gases such as nitrogen and oxygen can produce radioactive nuclides (e.g., Lal and Peters, 1967; Beer et al., 2012) including $^7$Be (half-life 53.22 days) and $^{10}$Be (half-life $1.4 \cdot 10^6$ years). They are called cosmogenic isotopes since Galactic and solar cosmic rays from the dominant source of these nuclides in the terrestrial system. Production rates of $^7$Be and $^{10}$Be vary in time following the intensity of Galactic cosmic rays (GCR) modulated by solar magnetic activity (the 11-year solar cycle), geomagnetic field strength, and also sporadic events of solar energetic particles (SEPs) (see, e.g., Usoskin and Kovaltsov, 2008; Kovaltsov and Usoskin, 2010).

Short-living cosmogenic isotope $^7$Be, which is relatively easy to measure via a proper collection system and $\gamma$-spectrometry, is routinely measured, with daily, weekly and monthly cadences, in near-ground air and precipitating water at different places around the globe in the framework of the atmospheric radiation monitoring. Concentrations of long-living $^{10}$Be are measured (via acceleration mass spectrometry, AMS) in natural stratified archives, typically in polar ice cores, with pseudo-annual or longer time resolution (e.g., Beer et al., 2012). In particular, $^{10}$Be recorded in polar ice cores serves as a main proxy for long-term solar-activity reconstructions in the past (e.g., Steinhilber et al., 2012; Usoskin, 2017; Wu et al., 2018).

In addition to the production pattern, defined by cosmic-ray fluxes, concentrations of beryllium isotopes at any location are also affected by atmospheric transport and deposition processes, which may essentially distort the production signal. Thanks to the short life of $^7$Be, high-cadence data, and broad distribution of the measuring sites, this isotope is often used as a tracer for atmospheric dynamics during the recent decades. Its atmospheric transport can be modelled with high accuracy using the known meteorological (wind) fields. On the other hand, data of $^{10}$Be are from the polar regions with low temporal resolution and extends far back in the past, to times when the meteorological data are not available. Accordingly, the atmospheric transport of $^{10}$Be is typically modelled using simplified box models based on a parametrization of some specific simulations (Heikkilä et al., 2013). Thus, there is a need for a realistic model able to simulate atmospheric transport and deposition of beryllium

focused in polar regions and (inter)annual time scales and potentially able to operate in a self-consistent mode, without the prescribed meteorology.

Since measurements of $^{10}$Be are laborious and expensive, their coverage for the last decades is low making it difficult to compare model results directly with measured data. However, since the two beryllium isotopes are believed to have similar transport/deposition being different only in the production and the lifetime, a model applied and verified for the $^{7}$Be transport can be generally applied also to $^{10}$Be. Here we develop such a model to trace isotopes of $^{7}$Be and $^{10}$Be in the atmosphere based on the chemistry-climate model SOCOL (SOlar Climate Ozone Links) v3, which has been specifically modified by including modules for the production, deposition, and transport of $^{7}$Be and $^{10}$Be.

The paper is organized as follows: Summary of the previous and existing models is presented in Section 2; Data of $^{7}$Be measurements is presented in Section 3; The model is described in great detail in Section 4; Obtained results are discussed in Section 5; Validation of the model by means of comparisons with measurements is presented in Section 6; Section 7 summarizes conclusions of the paper.

## 2  Summary previous and existing models

Studies of short-living $^{7}$Be isotope and its ratio to $^{10}$Be provide a precise way to probe large scale atmospheric dynamical properties and validate the corresponding atmospheric models (e.g., Raisbeck et al., 1981; Jordan et al., 2003; Ioannidou and Paatero, 2014). For the long-living $^{10}$Be isotope, different transport/deposition assumptions and models were used (e.g., McCracken, 2004; Heikkilä et al., 2009; Sukhodolov et al., 2017; Usoskin et al., 2020a). Here we review some key models.

Earlier simulation with the transport model GLOMAC1 by Brost et al. (1991) yielded the concentration and deposition of $^{7}$Be isotopes in agreement with the data within 20%. However, the authors conclude that the application of this transport model and scavenging scheme for simulations of deposition and concentration was not well suited for polar regions.

Later, simulations of $^{7}$Be and $^{10}$Be transport for present and future climatic conditions were performed by Land and Feichter (2003) with the general circulation model MA-ECHAM4. The model successfully captured the maximum $^{10}$Be/$^{7}$Be ratio in the lower polar stratosphere.

The general ability of a full atmospheric-dynamic model to reproduce the pattern of beryllium transport and deposition even at regional spatial and synoptic temporal scales was demonstrated by Usoskin et al. (2009a) who used the GISS model (Field et al., 2006) to simulate $^{7}$Be production and transport for two months of Jan – Feb 2005 and compared the model's results with daily-resolution near-ground $^{7}$Be air concentrations measured at eleven sites worldwide (see Table 1 in Usoskin et al., 2009a). The results implied that the model captures $^{7}$Be transport at synoptic and longer time scales (longer than 4 – 10 days, see Fig. 5 in Usoskin et al., 2009a) in both absolute levels and time variability. However, as noted by Koch et al. (2006), the GISS ModelE has a low bias (a factor of two) in the upper troposphere and lower stratosphere while agreeing well with the surface concentrations. It was proposed that this can be related to too low model production and that either the model delivers too much $^{7}$Be to the surface and/or does not scavenge efficiently enough. The GISS model was not further developed in this direction.

Modelling of beryllium isotopes in the Earth's atmosphere was performed earlier using the ECHAM5-HAM atmospheric model (Heikkilä et al., 2008a, 2009) and zonal mean production rates. Cosmogenic radionuclides $^{10}$Be and $^{7}$Be were modelled during the Maunder Minimum and the present-day climate (Heikkilä et al., 2008b). The ECHAM5-HAM simulates beryllium fluxes and concentrations in ice at Greenland sites well, in agreement with the data. In Antarctica however, the model has some difficulties in reproducing the very low level of precipitation which leads to an underestimation of the $^{10}$Be concentrations in ice. The precipitation rate modelled at Concordia site was a factor $2-3$ too high, leading to too low $^{10}$Be concentrations in ice, since, after full depletion of $^{10}$Be from atmosphere, further rain/snow dilutes its concentration in the ice. The $^{7}$Be surface air concentrations agree generally within a factor of two with the observed concentrations. However, modelled concentrations at two of Antarctic stations appeared too low. This model was very roughly parameterized (Heikkilä et al., 2013) being effectively reduced to a six-box model and not developed further in the application to beryllium-transport modelling.

A simplified 2D model of beryllium stratospheric sedimentation was developed by Delaygue et al. (2015) but also not supported further.

At present, even though a full modelling can be done, transport of $^{7}$Be is typically considered, for application purposes, with simple box-reservoir exchange models (e.g., Pacini et al., 2015; Zheng et al., 2021) or with air-mass tracing codes (Pacini et al., 2011; Brattich et al., 2020). The latter approach is applicable only to recent years where the wind field is known, moreover, it includes neither stratospheric dynamics nor depositional processes and, therefore, is not suitable for $^{10}$Be in polar ice.

Thus, several models of different complexity and accuracy have been developed in the recent past to model transport and deposition of $^{7}$Be and $^{10}$Be. However, due to the lack of support for the radionuclide simulations with ECHAM-HAM (Heikkilä et al., 2008b) and GISS (Field et al., 2006) climate models, a full working model capable of simulating production, transport, and deposition of $^{7}$Be and $^{10}$Be from all sources, is presently missing. The model should be 'user-friendly' (i.e., can be used by a user without a deep knowledge of climate modelling) and applicable for the past, when direct nudging to the meteorological re-analysis data is not possible.

Here we present a new development of the full chemistry-climate model (CCM) SOlar Climate Ozone Links (SOCOL) for modelling of production, transport, and deposition of the cosmogenic isotopes of beryllium as well as its validation with the available measurements of $^{7}$Be at high-latitude locations. The CCM SOCOL is potentially capable of simulation the beryllium transport and deposition and was already used (Sukhodolov et al., 2017; Usoskin et al., 2020a) for the estimation of $^{10}$Be deposition after major solar energetic particle (SEP) events, but it did not include the state-of-the-art interactive parameterization of the wet and dry deposition introduced recently (Feinberg et al., 2019). A recent model version SOCOL-AERv2, which simulates aerosols more realistically does not, however, include the treatment of all processes relevant to the beryllium life cycle and its applicability has not been evaluated. We have further upgraded the CCM SOCOL-AERv2 (Feinberg et al., 2019) here, by adding the production, transport, and deposition of $^{7}$Be and $^{10}$Be isotopes from both GCR and SEPs.

The main purpose of this work is to present a new combined model of beryllium production and transport and to confront its results with high-resolution (weekly) measurements of $^{7}$Be in near-ground air and precipitating water in polar regions. The isotope $^{7}$Be was chosen for the analysis because of the following reasons: (i) there is no systematic high-quality $^{10}$Be data in air samples since measurements of $^{10}$Be are much more difficult (AMS technology) then those of $^{7}$Be ($\gamma$-spectrometry); (ii)

**Table 1.** List of stations whose data were used for the present study. Data sources are: Finnish National Radiation and Nuclear Safety Authority (STUK); Radiation Protection Bureau of Health Canada (RPBHC); Complete Nuclear Test-Ban-Treaty Organisation (CTBTO).

| Station Name | Region | Location | Source | Time interval |
|---|---|---|---|---|
| Ivalo | Finland | $68^o39'09''$ N $27^o32'47''$ E | STUK | 2002 – 2008 |
| Rovaniemi | Finland | $66^o30'05''$ N $25^o44'05''$ E | STUK | 2002 – 2008 |
| Kajaani | Finland | $64^o13'30''$ N $27^o44'$ E | STUK | 2002 – 2008 |
| Kotka | Finland | $60^o28'$ N $26^o56'45''$ E | STUK | 2002 – 2008 |
| Yellowknife | Canada | $62^o26'32''$ N $114^o23'51''$ W | RPBHC | 2003 – 2008 |
| Kerguelen island | Port-aux-Français, France | $49^o21'$ S $70^o13'$ E | CTBTO | 2006 – 2008 |
| Punta Arenas | Chile | $53^o09'$ S $70^o55'$ W | CTBTO | 2006 – 2008 |

[10]Be measured in ice cores contains an additional component (deposition) which also affects the data and needs to be modelled separately. On the other hand, the model eventually aims at the modelling of the [10]Be isotope including its deposition. Since [10]Be is typically measured in polar ice cores with pseudo-annual or longer resolution, we are primarily interested to validate the model for high-latitude regions and (inter)annual time scales. The new model version SOCOL-AERv2-BEv1 has been developed here for systematic modelling of [7]Be and [10]Be production, transport and deposition in the atmosphere. This model set forms a methodological basis for a detailed study of the atmospheric dynamic and cosmogenic-isotope distribution in the terrestrial system. The model computations were performed with full nudging (using ERA5) to the observed meteorological data to verify the capability of the model to trace beryllium isotopes in the atmosphere. The model computations were directly compared with measurements of [7]Be isotopes in different high-latitude location to validate the model.

## 3   Data of [7]Be mesurements

For the model verification, we used data of the measured activity (concentration) of [7]Be isotope in near-ground air, with weekly sampling rate (corrected for the decay) performed in different hemispheres, for the period 2002 – 2008. Higher-cadence data are not necessary for several reasons: (i) Data availability, since weekly sampling is a standard resolution for most [7]Be-measuring station; (ii) Inability of full atmospheric-dynamics models to catch the sub-synoptic time scales because of the rough grid size (see Heikkilä et al., 2009; Usoskin et al., 2009b); (iii) we focus on longer time scales (annual and inter-annual) for the application to [10]Be in ice cores. On the other hand, rougher-resolution data (e.g., monthly or quarterly) are more uncertain because of the decay of short-living [7]Be. Thus, we found the weekly-resolution data optimal for our purposes.

Traditionally, for short-living isotopes, the measured quantity is not concentration but activity, in units of becquerel (number of radioactive decays per second) per a cubic meter of air (Bq/m$^3$), as directly measured by a $\gamma$-spectrometer. Correction for decay between the times of sample collection and measurement is standardly performed (see Leppänen et al., 2012). The activity and concentration can be easily translated to each other using the known lifetime of the isotope.

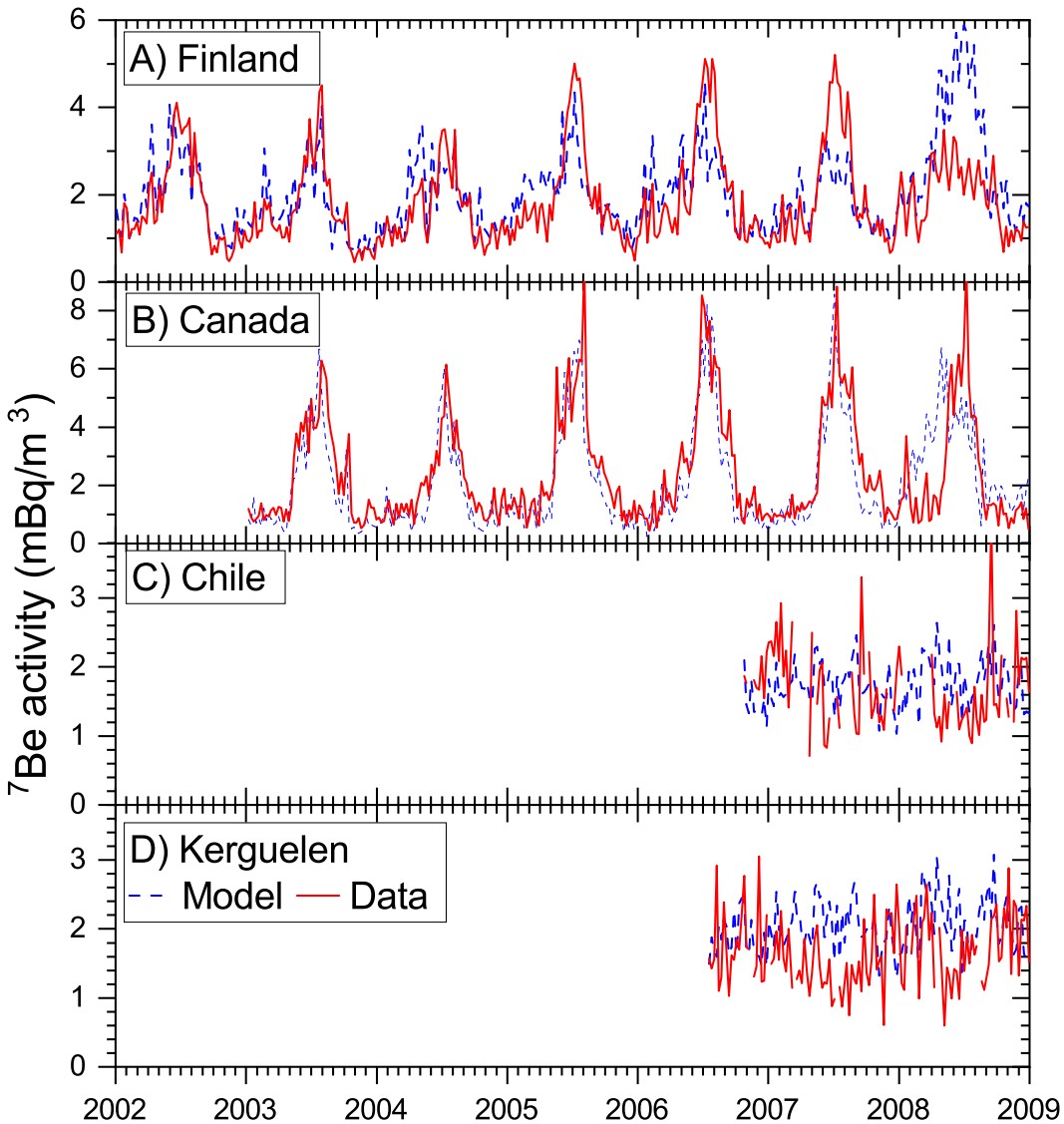

**Figure 1.** Datasets of $^7$Be activity in the near-ground air used in this study, including (panels A through D respectively) Finland (a composite of four stations), Yellowknife (Canada), Punta Arenas (Chile), and Kerguelen island (France) – see detailed information in Table 1. The measured and modelled activities are shown as solid red and dashed blue curves, respectively.

Here we use data from four nearly antipodal high-latitude locations (see Table 1). Two locations are in the Northern hemisphere: Finland (a set of stations) and Canada (Yellowknife); and two in the Southern hemisphere: Kerguelen island (French) and Punta Arenas (Chile). The longitudinal separation between the locations in each hemisphere is greater than 140°. The data series are shown in Figure 1 and described below.

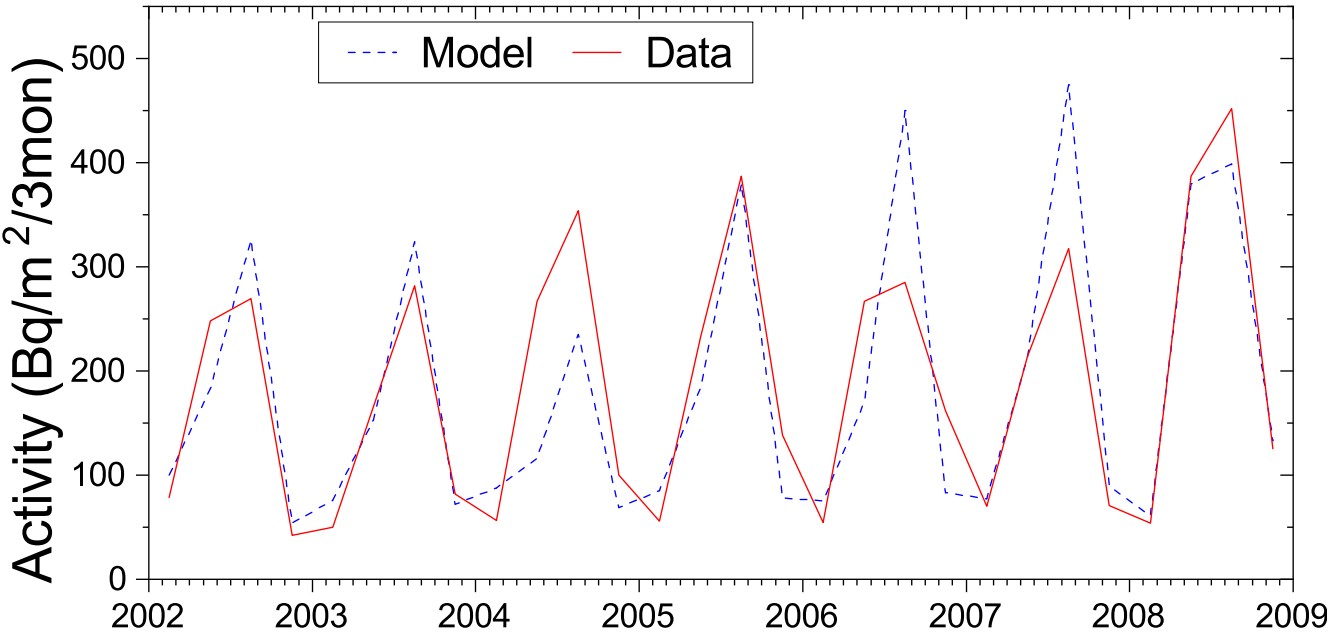

**Figure 2.** Quarterly total deposition of [7]Be in northern Finland stations (Ivalo and Rovaniemi) for the period 2002–2008. The blue dashed line represents the total modelled deposition, while the red line depicts the measured data.

Finnish data of the measured [7]Be activity were provided, for the period 2002–2008, by the Finnish National Radiation and Nuclear Safety Authority (STUK) for four sites (see Table 1): Ivalo, Rovaniemi, Kajaani and Kotka, spanning meridionally 8° in latitude. The isotope of [7]Be was collected on air filters using a high-volume air sampler on a weekly basis, and then its activity was measured with an HPGe (High Purity Germanium) detector inside a 4-inch thick lead shielding in the STUK laboratory. The statistical measurement uncertainty is 3–5% (see a detailed description of the sampling and measurement
procedure in Leppänen et al., 2012). The data from the four stations were combined as the average of the four locations for each time interval, to represent a regional-scale dataset, as shown in Figure 1A.

  We also used [7]Be data from the Yellowknife site (Canada), see Figure 1B, kindly provided by the Radiation Protection Bureau of Health Canada. The original data are with daily cadence but have been combined into weekly samples (correction for decay applied) for consistency with other datasets. The total uncertainty for the measured [7]Be activity is defined mostly by
155 statistics and is approximately 7–8%.

  For the Southern hemisphere, we considered weekly [7]Be activities measured at Punta Arenas (Chile, CTBTO station code CLP18). During 2002–2006, the data contain many long breaks (about half of weekly data are missing), hence we considered data since October 2006 only, as shown in Figure 1C.

  Another Southern Hemisphere station is located at Port-aux-Français, Kerguelen Island (CTBT station code FRP30) with
160 data available since 2006 (Figure 1D). The data for Punta Arenas and Kerguelen Island were obtained via the virtual Data

Exploration Centre (vDEC) of the Preparatory Commission for the Complete Nuclear Test-Ban-Treaty Organisation (CTBTO). A full description of the sampling and measurement of the CTBTO data is available elsewhere (Miley et al., 1998; Medici, 2001).

In addition to $^7$Be activity measured in near-ground air, we also exploited data on the quarterly total-deposition $^7$Be measurements (without separating dry and wet depositions) measured in collected precipitation water at Rovaniemi and Ivalo stations, as provided by STUK. The fallout sample collection period was one month, but the three samples were combined together to form a quarterly sample. Hence, the four measured samples for each year cover period of January to March, April to June, July to September and October to December, respectively. The reference date was set to the middle of the sampling period (since the decay time of $^7$Be is shorter than the sampling interval) and the activities were decay-corrected to this date (see Leppänen, 2019). The $^7$Be deposition averaged over the two locations is shown in Figure 2.

## 4 Model description

We used an extended version of the CCM SOCOLv3 (Stenke et al., 2013) with the aerosol module - SOCOL-AERv2 (Feinberg et al., 2019). This version consists of the general circulation model MA-ECHAM5 (Hommel et al., 2011), the atmospheric chemistry module MEZON (Egorova et al., 2003) combined with an aerosol module AER (Weisenstein et al., 1997), interacting with each other every two modelling hours. In this study, MA-ECHAM5 is nudged with fifth generation ECMWF atmospheric reanalysis data of the global climate ERA5 (eraiaT42L39) reanalyses. The chemical part of the MEZON core includes 56 chemical species of oxygen, hydrogen, nitrogen, carbon, chlorine, bromine, and sulfur groups, 295 gas-phase reactions, 64 photolysis reactions, and 16 heterogeneous reactions on stratospheric aqueous sulfuric acid aerosols as well as three types of polar stratospheric clouds. SOCOL uses the horizontal resolution T42, where $T$ refers to triangular truncation and the number denotes the series of spherical harmonics. The horizontal resolution is about $2.8° \times 2.8°$ ($\approx 300 \times 300$ km in the equatorial region). The vertical grid is defined using a sigma-hybrid coordinate system, which is calculated by a linear combination of surface pressure and constant coefficients that define the vertical coordinate. The model has 39 vertical levels between the Earth's surface and the 0.01 hPa level ($\approx$80 km). The real orography is smoothed over the model grid cells. The aerosol module exploits a sectional microphysical scheme where the aerosol particle size distribution is represented by 40 discrete size bins. The model simulates the advective transport of stratospheric aerosol by the Brewer-Dobson large-scale circulation (Butchart, 2014) and gravitational sedimentation. We have made some adaptations of the main model to focus on beryllium transport, as describes below.

### 4.1 Production rates of $^7$Be

Production of both $^7$Be and $^{10}$Be was computed using the CRAC:Be (Cosmic-Ray Atmospheric Cascade: application to Beryllium) model (Poluianov et al., 2016) which is the most recent and accurate model of cosmogenic isotope production. This model simulates, using the GEANT4 Monte-Carlo simulation tool (Geant4 collaboration, 2020), the full nucleonic-muon-electromagnetic cascade, induced by primary cosmic-ray particles in the atmosphere. This model is an essential upgrade of

the previous versions of the CRAC:Be model (Usoskin and Kovaltsov, 2008; Kovaltsov and Usoskin, 2010) based on the CORSIKA simulation tool (Heck et al., 1998). The CRAC model presents a set of accurately computed yield functions of production of cosmogenic isotopes for different primary particle's types, energy and atmospheric depths. By a numerical integration, above the local geomagnetic rigidity cutoff (Cooke et al., 1991), of the product of the yield function and the known (or assumed) energy spectrum of primary cosmic rays (of either GCR or SEP origin), one can obtain the isotope's production rate as a function of location (via the geomagnetic rigidity cutoff), atmospheric height (via the yield function) and time (via the time-variable energy spectrum). These production rates, computed for each location and with hourly temporal cadence, were used as an input for the SOCOL model, providing a 3D+time source of the $^7$Be and $^{10}$Be in the atmosphere.

Production due to Galactic cosmic rays (GCR) which always bombard the Earth with slightly variable intensity due to solar modulation was computed by applying the force-field model (Gleeson and Axford, 1968) with the modulation potential reconstructed from the world neutron-monitor network (Usoskin et al., 2017). Geomagnetic field was taken according to the International Geomagnetic Reference Field model (IGRF, Thébault et al., 2015) in the eccentric dipole approximation (Fraser-Smith, 1987; Usoskin et al., 2010). While GCR are always present near Earth, sporadic solar particle events (SPEs) take place occasionally (Vainio et al., 2009; Desai and Giacalone, 2016) and can be sufficiently strong to be recorded by ground-based neutron monitors (Raukunen et al., 2018; Usoskin et al., 2020b). A large number of cosmogenic isotopes can be produced during such events (Usoskin et al., 2020a), which can be several orders of magnitude stronger on the multi-millennial time scale than for the last decades covered by direct observations, with strong signals in the polar $^{10}$Be concentrations (e.g., Sukhodolov et al., 2017). To model the contribution of severe SPEs, we modelled the production of beryllium by SPE of 20-Jan-2005 (GLE #69), which was the strongest event during the studied period (1996 − 2008) and the second strongest directly observed (1951 − 2020). The same yield function and geomagnetic cutoff approach, as for GCR, were applied along with the SEP energy spectrum obtained for this event by Usoskin et al. (2020b). To study the possible effect of seasonality on the $^7$Be and $^{10}$Be transport we simulated the same SEP event as if it occurred not only on its actual date in mid-winter (20-Jan-2005) but also mid-spring, summer, and autumn (20-Apr, 20-Jul and 20-Oct of 2005, respectively).

Figure 3 shows the zonal mean production of $^7$Be as a function of the atmospheric pressure and northern geographical latitude from two sources: daily GCR (panel A) and SPE (panel B). Since the duration of a strong SPE takes several hours, we compare it with the daily production of $^7$Be by GCR. One can see that the polar stratospheric production due to SPE grossly overwhelms that by the daily GCR flux, but it fades towards tropical latitudes. Since the production of $^7$Be is nearly symmetric between the global hemispheres, as defined by the geomagnetic shielding, only the Northern hemisphere is shown. Even though the production rate is significantly higher in the polar region, its contribution to the global production is not dominant, because of the small area of the polar regions. For the GCR-related production, the dominant production region is located in the mid-high latitude (40 − 70°) stratosphere and upper troposphere (above 400-hPa level or ≈7 km). Production of $^7$Be by SEPs is confined to the polar (latitude >55°) upper stratosphere and mesosphere (above 50-hPa level or ≈35 km). Hardly any beryllium nuclei are produced by SEPs in the tropical region. In all cases, the maximum of production lies at mid-/high-latitudes in the stratosphere, and the production decreases towards the surface, because of the increasing atmospheric depth, and equatorward due to the geomagnetic shielding. Because of the much softer energy spectrum, SEPs produce $^7$Be at

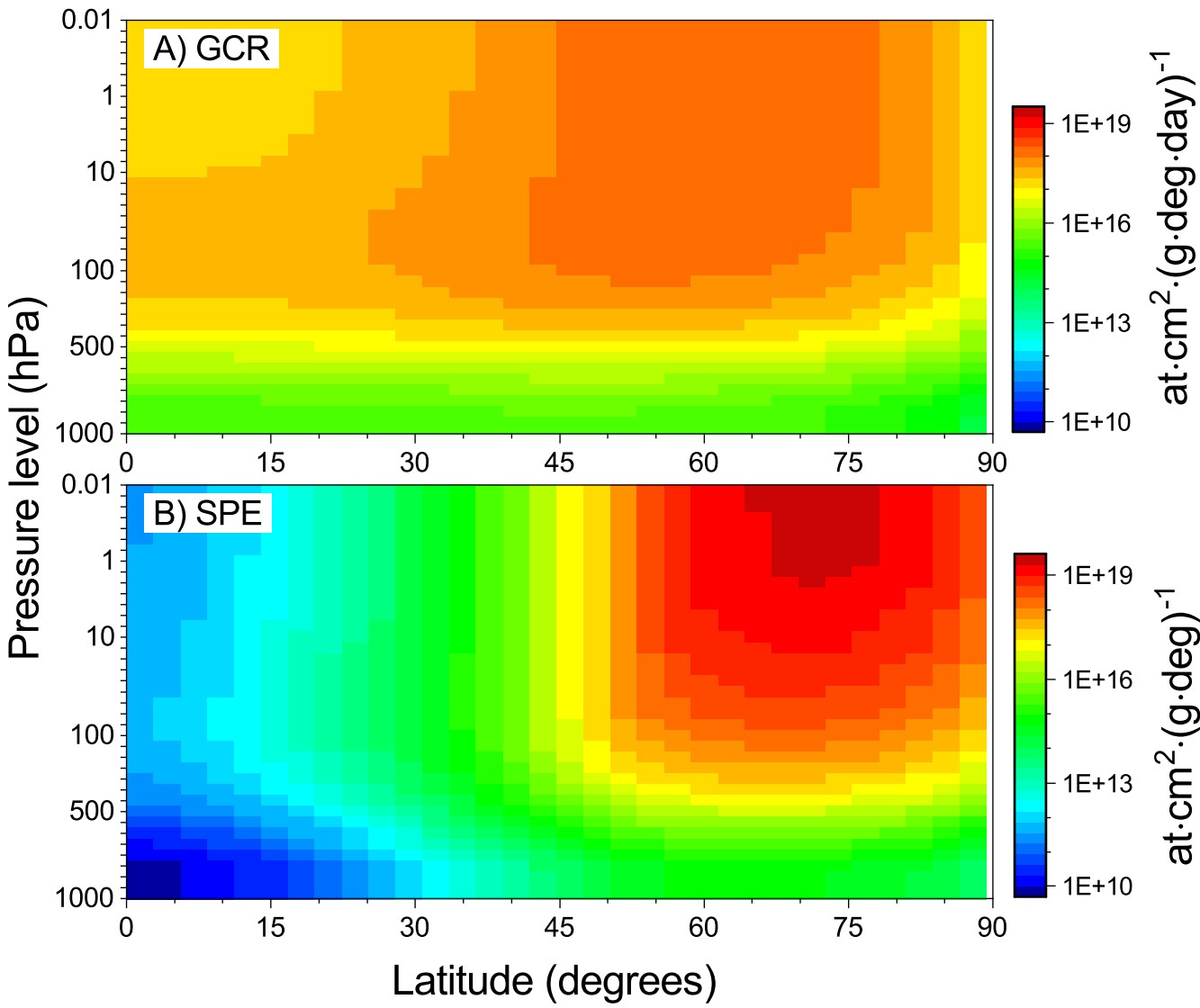

**Figure 3.** Zonal mean production of $^7$Be in the Northern hemisphere as a function of geographical latitude and model's atmospheric pressure level. Panel A: daily production by GCR averaged over the period of 2002 – 2008. The color scale (on the right) is given in units of atoms per day per degree of latitude per g/cm$^2$. Panel B: Total production by SPE of 20-Jan-2005. The color scale is in units of atoms per degree of latitude per g/cm$^2$.

shallower atmospheric depths and higher latitudes than GCR do. The production pattern for $^{10}$Be is similar to that for $^7$Be but the absolute values are slightly smaller.

## 4.2 Decay of $^7$Be

$^7$Be is a short-living isotope whose decay cannot be neglected in comparison with the typical transport/deposition time, especially in the stratosphere, in contrast to long-living $^{10}$Be isotope. Accordingly, a standard decay probability of 0.054% per hour (corresponding to the isotope's mean half-life of 53.22 days) was applied during tracing of $^7$Be in a way similar to that used by Golubenko et al. (2020) for $^{222}$Rn. No decay needs to be applied to $^{10}$Be for this kind of modelling. Figure 4 shows the modelled global content of the two beryllium isotopes in the atmosphere after an instant production by the SPE event of 20-Jan-2005 (day zero). One can see that the amount of $^{10}$Be decreases nearly exponentially reflecting a slow removal of the isotope through precipitation. A typical time of the isotope removal is $\tau_{10\text{Be}} = 550 \pm 100$ days, implying the residence time of a few years (cf., e.g., Beer et al., 2012). This time is comparable to the $1-2$-year atmospheric residence time of other isotopes (e.g., $^{137}$Cs and $^{241}$Am) defined after nuclear-weapon tests (Haltia et al., 2021). The decrease of $^7$Be isotope concentration is much faster and nearly perfectly exponential with $\tau_{7\text{Be}} = 72 \pm 3$ days, which includes both decay and the removal. Since the removal process is assumed to be identical for both isotopes, the concentrations of $^{10}$Be can be used to correct the $^7$Be decay for the system effects. The decay time of $^7$Be, estimated in this way, appears $1/\tau = 1/\tau_{7\text{Be}} - 1/\tau_{10\text{Be}}$ as $\tau = 82 \pm 4$ days which is generally consistent with the expected e-folding lifetime of $^7$Be of 77 days, implying that the decay was accounted correctly in the model. Thus, the $^7$Be isotope is fully removed from the atmosphere, mostly due to decay, within several months.

## 4.3 Transport of cosmogenic isotopes $^7$Be and $^{10}$Be

Beryllium atoms are captured by different aerosols in the air that can enhance or suppress its transport by gravitational sedimentation. Since the typical size of aerosol particles is less than 0.2 $\mu$m (Pierce et al., 2010), with a modal value of 0.05 $\mu$m, the corresponding gravitational sedimentation velocity ranges between 200 and 1000 m yr$^{-1}$ and does not play an important role in the lower stratosphere, where it is much smaller than the vertical wind speed (e.g., Weisenstein et al., 2015; Zhou et al., 2006). The role of gravitational sedimentation was evaluated by Delaygue et al. (2015) using numerical experiments with a 2D model, which includes detailed sulfate aerosol microphysics. By comparing no-gravitational-sedimentation and reference model runs, they concluded that "... sedimentation of $^7$Be atoms does not seem to play an important role in removing $^7$Be out of the stratosphere, because radioactive decay and transport are faster processes". The marginal importance of gravitational settling and identical transport of isotopes in gas and aerosol forms was also described in the seminal work of Lal and Peters (1967). It should be noted that we treat $^7$Be as gas only for the advective transport when the result is the same for the small particles and gas components.

Different processes such as stratospheric mixing, stratosphere–troposphere exchange, tropospheric transport and deposition, are realistically modelled by the CCM SOCOL (Feinberg et al., 2019). In this study, the advective transport of $^7$Be and $^{10}$Be in gas form was performed using Flux-Form Semi-Lagrangian Transport Schemes (Lin and Rood, 1996) embedded in ECHAM5.

Beryllium is also transported vertically by deep, shallow, and mid-level convective motions. The transport is defined by the convective mass fluxes of the standard grid-box ECHAM5 convection scheme described by Tiedtke (1989) and Nordeng

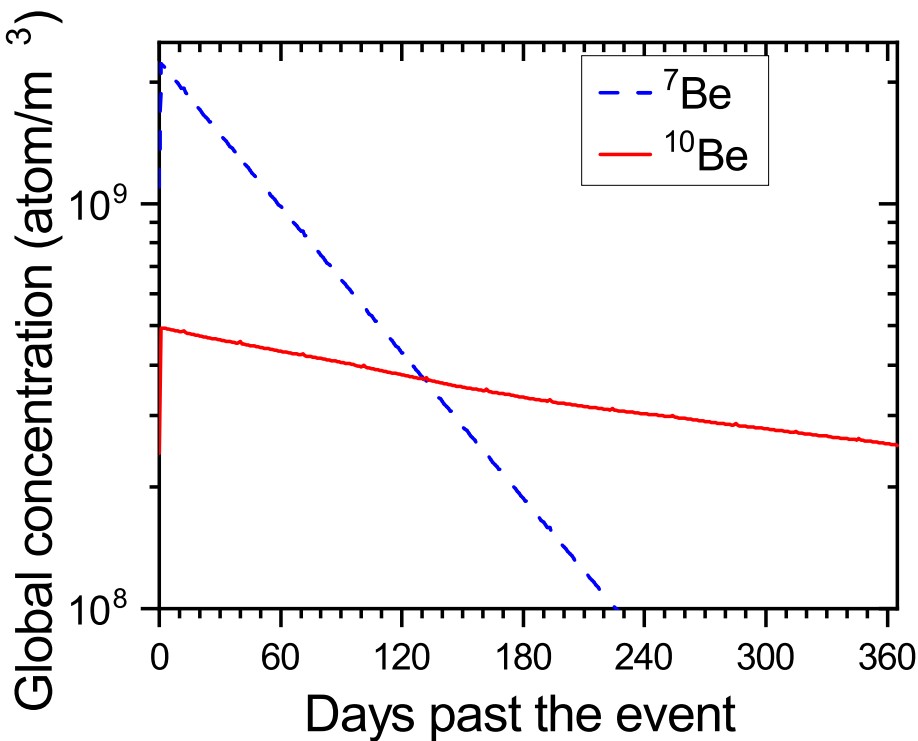

**Figure 4.** Computed global mean concentration of two beryllium isotopes, $^7$Be and $^{10}$Be, produced by SPE of 20-Jan-2005 (day zero) as function of time. The nearly exponential decay of concentration ($\tau = 550\pm100$ days and $72\pm2$ days for $^{10}$Be and $^7$Be, respectively) includes both decay and sink (deposition) of the isotopes.

(1994). The transport parameterization by mixing in the boundary layer is based on the eddy diffusion method and the vertical turbulent fluxes are related to the gradient of the considered quantity (Roeckner, 2003).

### 4.4 Deposition of the $^7$Be and $^{10}$Be

Dry deposition is typically modelled using a simplified approach that assumes constant dry deposition velocities over land and ocean, without accounting for seasonal or geographical variability. The tropospheric washout of gases can be calculated by using a constant removal rate, irrespective of precipitation occurrence (Hauglustaine et al., 1994). SOCOLv.3 employs a more sophisticated scheme based on the surface resistance approach for the estimation of dry deposition velocities (Wesely, 1989). Deposition of beryllium isotopes is parameterized as a function of surface properties, solubility, and reactivity of the considered

species (Kerkweg et al., 2006a). These properties for the modelled beryllium are identical to those for small sulfate aerosols. It means that beryllium atoms are considered as gas molecules only for the transport process, while with respect to the dry and wet depositions, they are treated as sulfates. The applied dry deposition scheme also considers the actual meteorological conditions and different surface types.

The interactive wet deposition scheme used in CCM SOCOLv.3 exploits the EAYSY2 version of the Scavenging (SCAV) submodule in the ECHAM/MESSy Atmospheric Chemistry (EMAC) model. Detailed description of the interactive wet-deposition scheme is presented and discussed elsewhere (Kerkweg et al., 2006b; Tost et al., 2007; Tost et al., 2010). The parameterization is based on the model-generated available liquid water in (cloud water content) and below (precipitating water) clouds and uptake/release form droplets, which depends on the concentration and solubility of the considered species. Scavenging coefficients for gas-phase species are calculated based on Henry's law equilibrium constants. The Henry constants for beryllium are considered identical to those of small sulfate aerosols.

## 4.5 Setup of numerical experiments

Since [10]Be is removed from the stratosphere via the stratosphere-troposphere exchange and the stratospheric mixing time is long (a few years), we have performed a 6-year spin-up of the model for the period $1996-2001$ to allow [10]Be to reach equilibrium conditions. For [7]Be, such a long spin-up is not needed since it is mostly removed from the atmosphere by decay, but we kept it similar to [10]Be for consistency. After the spin-up period, we initiated a 7-year $(2002-2008)$ run with full nudging (a linear relaxation of thermodynamic parameters: temperature, divergence, and vorticity of the wind field). In this way, we can validate the model by comparing the results with the measurements.

## 5 Modelling results

## 5.1 Global patterns of [7]Be transport and deposition

Figure 5 shows the zonal mean concentration of [7]Be averaged over the Boreal summer and winter times for $2002-2008$, as modelled here. While the isotope production is symmetric between the hemispheres, the concentrations are not completely symmetric and depend on the large-scale circulation patterns, which are different in the summer and winter hemispheres. The stratospheric meridional transport is driven by forces from the breaking planetary, synoptic-scale, and gravity waves. Stratospheric air ascends inside the so-called tropical pipe and then moves poleward descending at middle and high latitudes, predominantly in the winter hemisphere, where a more intensive stratosphere-troposphere exchange leads to slightly smaller concentrations of the [7]Be in the stratosphere during the cold seasons. Lower-stratospheric air descends relatively fast over the middle latitudes where the stratosphere-troposphere exchange is fast, while at high latitudes it resides longer in the lower stratosphere (Cohen et al., 2014).

Figure 6 presents distributions of the total (wet and dry) deposition of [7]Be averaged over $2002-2008$. The deposition is low in arid regions, e.g., the Sahara and the Middle East. It is also low to the West of the continents, following the low precipitation above the cold ocean currents. The precipitation rate is the highest in the tropical convection zone due to the highest temperature gradients and strong convection in the equatorial area. The deposition, however, is not very high at low latitudes because both the production rate and the downward transport of [7]Be are low in the tropics. It is important that gradients of the deposition

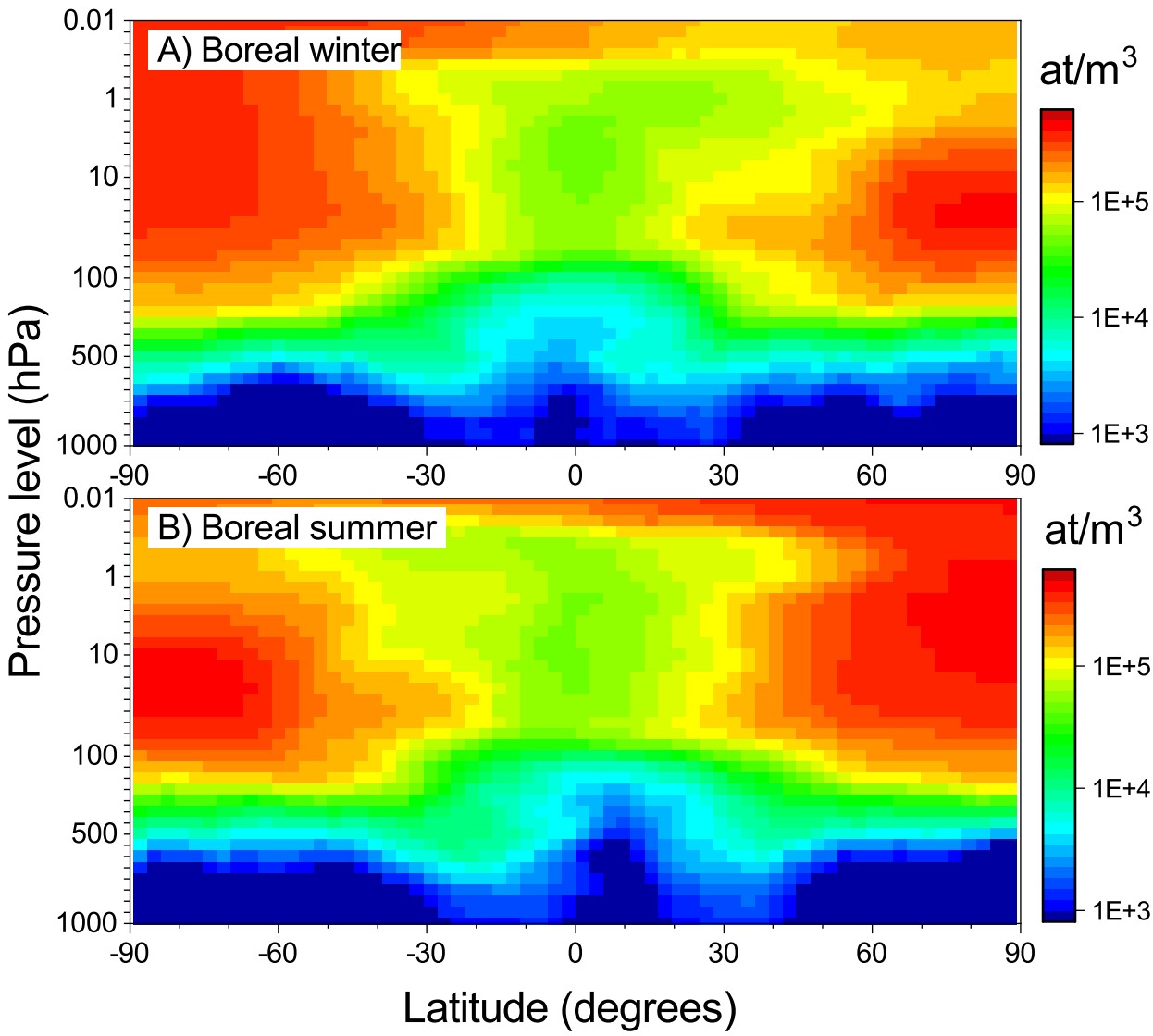

**Figure 5.** Modelled zonal mean concentrations of [7]Be during the Boreal winter (panel A) and Boreal summer (panel B) seasons averaged over 2002 – 2008.

can be strong even on the regional scale, suggesting that meteorological processes can strongly influence beryllium deposition at any given location (e.g., Usoskin et al., 2009b). This pattern agrees well with a similar previous study (Heikkilä et al., 2009).

### 5.2 The effect of a strong SPE

While GCR always bombard the Earth's atmosphere with slightly variable intensity, major SPEs make a different impact, producing sporadic short (typically several hours) but very intense fluxes of energetic particles entering the atmosphere mostly

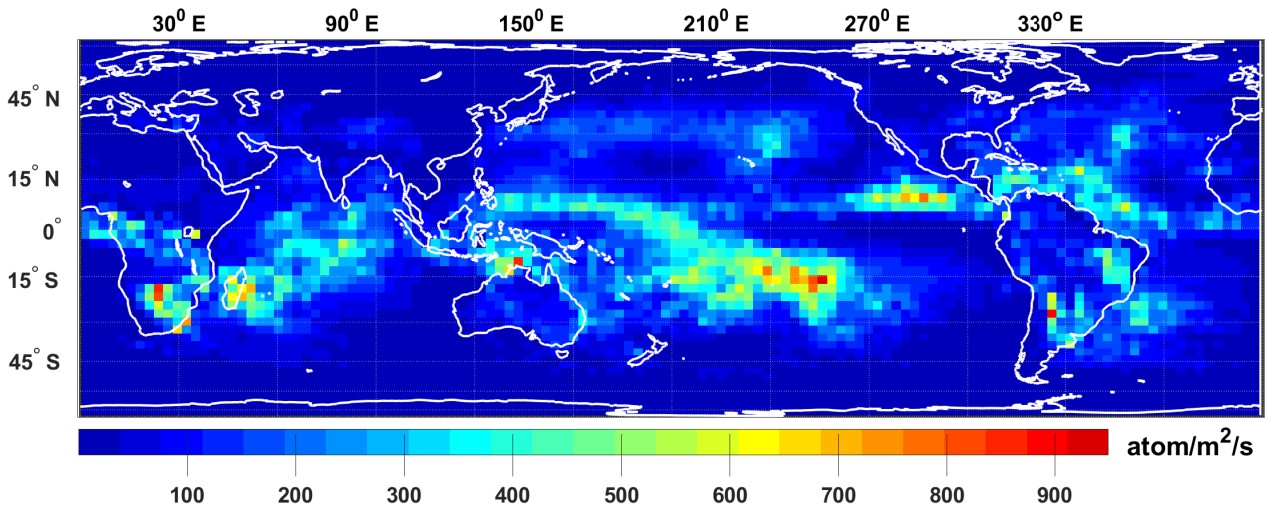

**Figure 6.** Lambert Equal-Area Cylindrical Projection map of $^7$Be total (wet and dry) deposition averaged over the period of $2002 - 2008$.

in polar regions. Potentially, extreme SPEs, orders of magnitude stronger than those observed during the recent decades, can be recorded in cosmogenic isotope data (Usoskin et al., 2006; Usoskin et al., 2020a), and a proper model is needed to study them (Sukhodolov et al., 2017). Here we modelled transport of such SPE-produced beryllium based on the strong SPE of 20-Jan-2005. In order to distinguish between different sources, we traced five beryllium tracers for the $^7$Be isotope: one for GCR-produced beryllium and four tracers for the SEP-scenarios as described above.

Production of the $^7$Be isotope during this SPE (Figure 3B) appears mostly in the polar stratosphere and lower mesosphere. After production, the isotope starts decaying and being transported by the air dynamic. Figure 7 shows an example of the SPE-produced isotope concentration on the $30^{t}h$ day after the event (19-Feb-2005). One can see that the production pattern is smeared by the transport, leading, in particular to an essential hemispheric difference. It is interesting that tropospheric concentrations are higher at mid/low latitudes of about $30°$ than in polar regions, because of the atmospheric circulation. As an example, the modelled activity of $^7$Be in near-ground air is shown in Figure 8 as averaged over Finland. The locally (polar troposphere) produced beryllium dominates during the first 20 days after the event, but then the transport starts playing a role, leading to the very low concentration during the subsequent period. The level of activity for this event in the near-air in Finland is very low, a factor 30-100 times lower than the typical level of activity due to GCR during the winter season (see Figure 1A).

Although the SPE of 20-Jan-2005 was a very strong one, the second strongest directly observed, its imprint in cosmogenic isotopes are not detectable on the background of air-transport dynamic and GCR variability. An SPE must be stronger by a factor of ten or more than this one to become detectable in isotope records even on the daily scale (cf. Sukhodolov et al., 2017; Usoskin et al., 2020a).

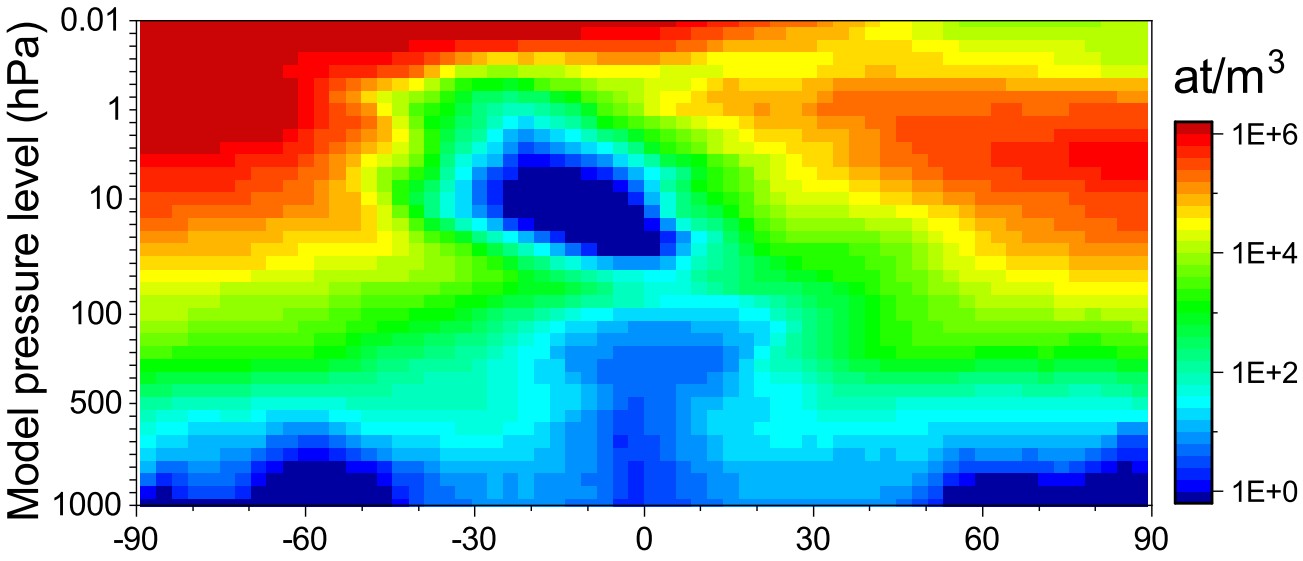

**Figure 7.** Distribution of the concentrations of $^7$Be produced by the SPE of 20-Jan-2005, on 19-Feb-2005, viz. 30 days after the event.

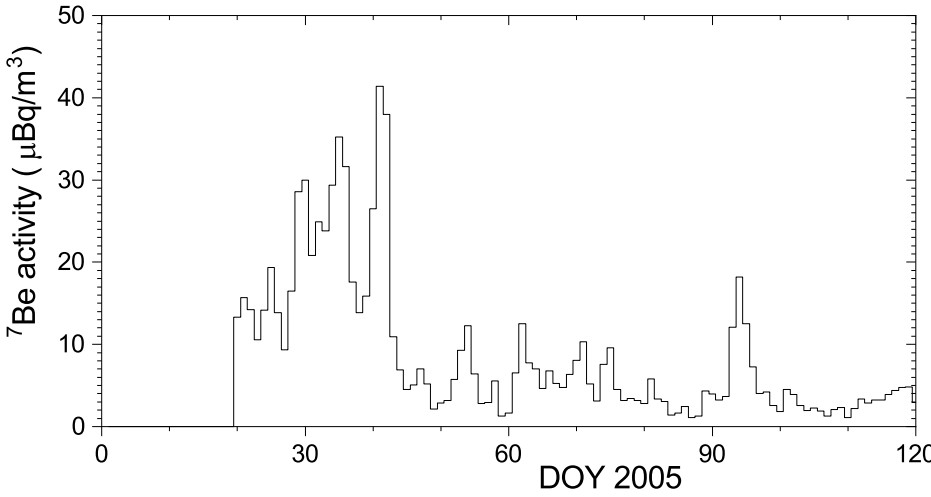

**Figure 8.** Modelled activity of $^7$Be, produced by the SPE of 20-Jan-2005, in near-ground air in Finland.

The transport of $^7$Be and $^{10}$Be after production may differ for different seasons. In order to study that, we simulated the same SPE of 20-Jan-2005 as if it occurred in mid-spring (20-Apr-2005), mid-summer (20-Jul-2005) and mid-autumn (20-Oct-2005). Figure 9 shows the concentration of $^{10}$Be in the northern hemisphere (85° latitude) for these cases. Here we consider $^{10}$Be because of the fast decay of $^7$Be which smears the time variability of the concentration. The most pronounced and long-lasting effect occurs from the SPE taking place during the autumn (panel d) when a strong concentration of $^{10}$Be persists in

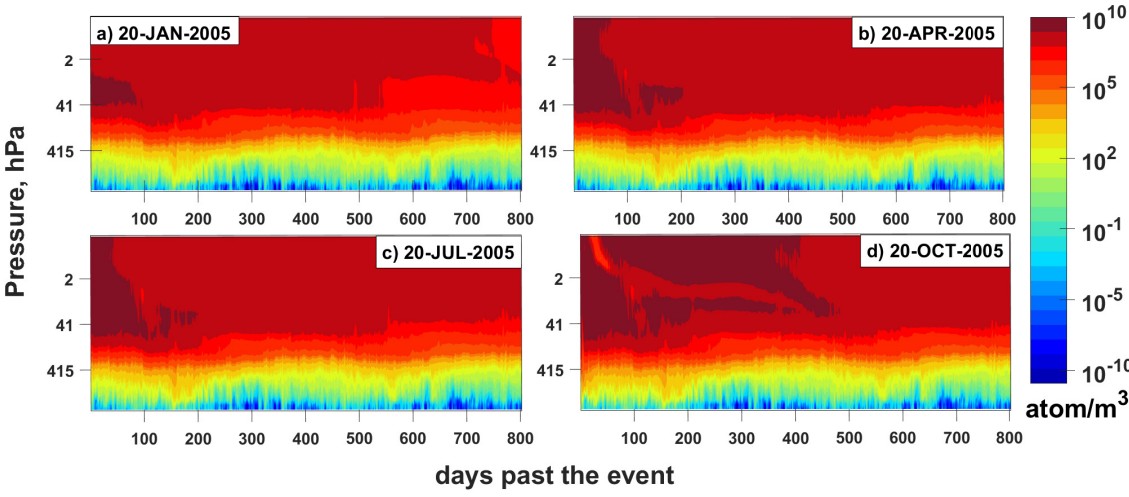

**Figure 9.** Polar (at 85° N latitute) mean concentration of $^{10}$Be in northern hemisphere for SPE occurred in different seasons.

the stratospheric region for about one year and gradually decreases after that for a few more years. The patterns of the SPE occurring during summer (panel c) and spring (panel b) seasons are similar to each other in that it has a shorter residence time (only a few months) and a smaller spread of $^{10}$Be. The winter scenario suggests somewhat faster removal of $^{10}$Be from the atmosphere, but it also stays in the upper atmosphere for several years. The concentration in near-ground air is mostly defined by the seasonal cycle and is consistent between different scenarios of the SPE occurrence date.

## 6 Evaluation of the model by comparison with direct $^{7}$Be measurements

In this section, we validate the model by comparing the simulation results with actual measurements of the near-ground-air $^{7}$Be concentration in air filters performed in Boreal and Austral high-latitude regions (Section 3), as shown in Figure 1.

First, we compared the modelled $^{7}$Be activity with the measured one for the all-Finland record, compiled from four stations (see Section 3) as shown in Figure 1A. The linear Pearson correlation between the modelled and measured weekly series is highly significant $r$=0.68±0.02 ($p$-value $<10^{-6}$). The standard correlation analysis does not distinguish between different frequencies/time scale. Therefore, we show in Figure 10 the global wavelet coherence, which is an analogue of the correlation coefficient but extending it into the frequency domain, between the modelled and measured $^{7}$Be activities.

Figure 10 depicts the coherence for the Finnish stations. The coherence is highly significant (confidence level above 95%) at all time scales between one month and two years. Although there are discrepancies on the very short time scales, related to the synoptic noise, the model correctly reproduces some strong spikes in the beryllium activity, as e.g., in the earlier 2003, which

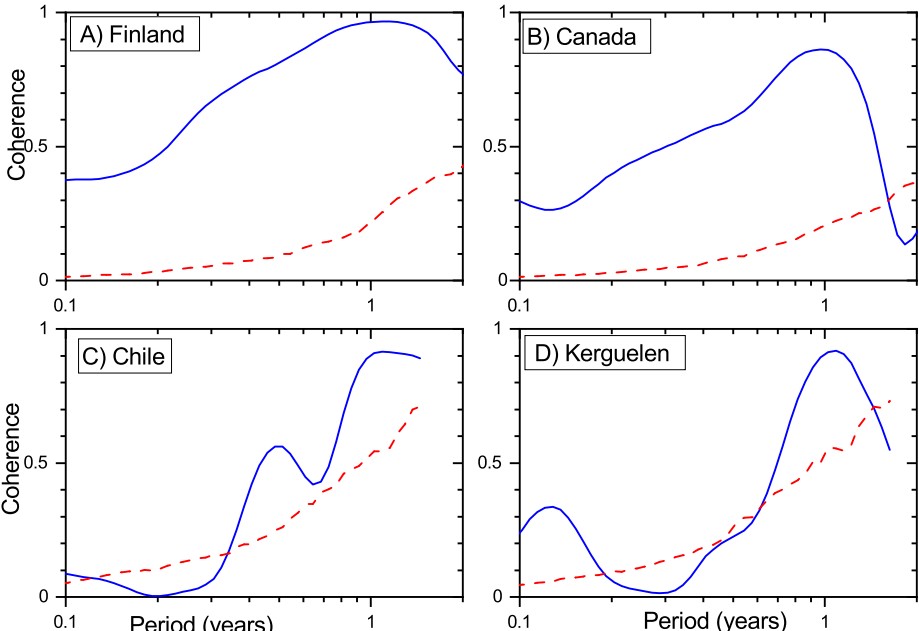

**Figure 10.** Global wavelet coherence (Morlet base, $k = 3$) between modelled and measured [7]Be activity for the four locations considered here. The red dashed line denotes the 95% confidence level against the AR(1) red noise.

are often related to sudden stratospheric warming (SSW) events (Pacini et al., 2015; Brattich et al., 2021). It is important that the long-term variability (at annual time scale) is correctly reproduced by the model.

However, neither correlation coefficient nor coherence analyses provide information on the absolute level match between the series. In order to assess possible model biases we considered the difference between the modelled and measured weekly [7]Be concentrations as shown in Figure 11A. The difference is insignificant $0.16 \pm 0.7$ mBq/m$^3$ implying the absence of a systematic bias in the model. The mean (averaged over seven yearly values) level of the modelled activity (2.05 mBq/m$^3$) is close to the measured one (1.86 mBq/m$^3$), within 10% (the difference is $0.19 \pm 0.19$ mBq/m$^3$). The $z$-test suggests that this difference

is insignificant. On the other hand, there is an essential difference between the model and the measurements regarding the magnitude of summer peaks in 2007 and 2008. It could be potentially related to the local atmospheric aerosol properties, but the AERONET (AErosol RObotic NETwork) (Emery and Camps, 2017) station in Sodankylä (Northern Finland) did not show any anomalies for that time.

For the Canadian station, the agreement between the modelled and measured activities (Figure 1B) is nearly perfect, in-

cluding a correct reproduction of the very strong seasonal cycle and sudden short spikes, as, e.g., in late 2003. The Pearson correlation coefficient is $r = 0.78 \pm 0.02$ ($p$-value $< 10^{-6}$). The coherence between the simulated and measured datasets (Figure 10B) is highly significant at all time scales between one month and 1.5 years. The difference between the modelled and measured concentrations is $-0.25 \pm 1.05$ mBq/m$^3$ (Figure 11B). The overall mean level of modelled activity (2.15 mBq/m$^3$) is close to the measured one (2.4 mBq/m$^3$), within 10% (the difference is $0.24 \pm 0.22$ mBq/m$^3$). The $z$-test suggests that the

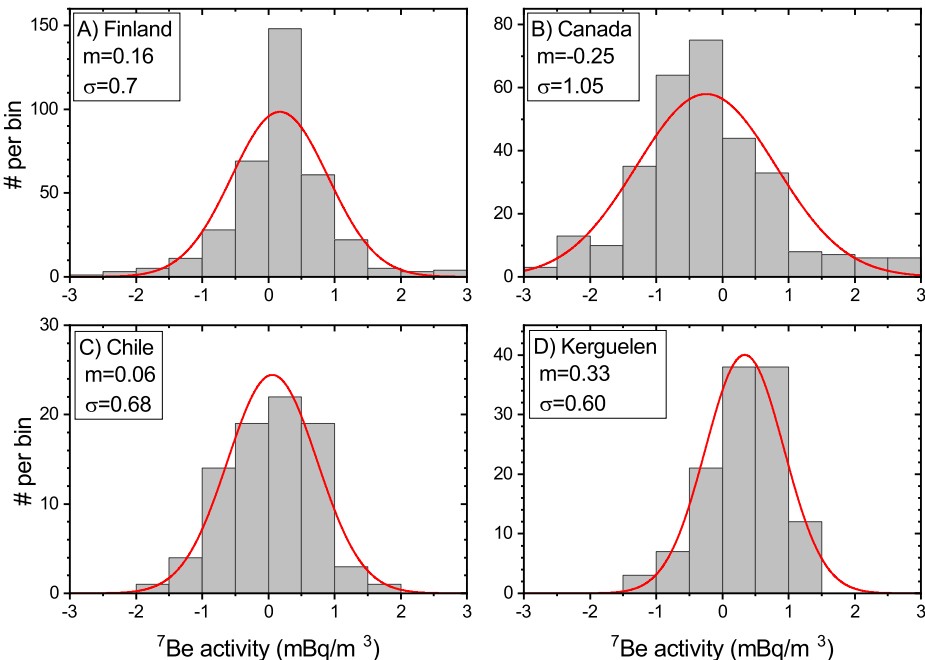

**Figure 11.** Histograms of the difference between modelled and measured [7]Be weekly concentrations (blue and red curves in Figure 1, respectively) for the four locations analyzed here (see Table 1). Red curves depict the best-fit Gaussians whose parameters (the mean $m$ and standard deviation $\sigma$) are quantified in the legends.

difference is insignificant, implying the absence of the model bias. The most pronounced difference between the model and the measurements was observed in 2008.

The Chilean data (Figure 1C) depicts a reasonable agreement between the model and the measurements. Since the data contains no seasonal pattern, as correctly reproduced by the model, the formal correlation is insignificant $r$ =0.06. However, it is dominated by the short-term variability, while the wavelet coherence analysis (Figure 10C) suggests that the two datasets are significantly coherent at time scales longer than 3 months. The difference between the modelled and measured concentrations is $0.06 \pm 0.68$ mBq/m$^3$ (Figure 11C). The mean levels of the modelled and measured series agree nearly perfectly, within 2%, viz. 1.74 vs. 1.71 mBq/m$^3$ for the modelled and measured activities, respectively, implying the absence of bias.

Data from the Kerguelen island (Figure 1D) also depicts a reasonable agreement, including correctly reproduced spikes and the lack of annual cycle. Similar to the Chilean data, the formal correlation is insignificant $r$ =0.06, but the coherence (Figure 10D) is highly significant at monthly and annual time scales. The difference between the modelled and measured concentrations is $0.33 \pm 0.6$ mBq/m$^3$ (Figure 11D). The mean levels of the modelled and measured series are reasonably matched, with the difference of 18%, viz. 2.02 vs. 1.65 mBq/m$^3$ for the modelled and measured activities, respectively, also indicating no significant bias. The observed small systematic discrepancy may be related to the peculiarity of this site which is

located on a small island (about 100 km across) in the middle of the ocean, while the model grid (about 300×300 km) may be too rough to catch the actual orography.

Therefore, we conclude that the model correctly (within 10%) calculates the mean levels of beryllium concentration for both northern and southern high-latitude locations, and contains no biases, at least for the locations where the orographic scale is comparable to the model's grid size but may have larger uncertainties for unresolvable spatial scales. It is important that the model correctly reproduces the seasonal variability or its absence as well as some sudden spikes, particularly during the late winter and spring seasons likely related to SSW events.

In addition to the concentration of $^7$Be in the near-surface air (expressed in measured activity), we also compared the $^7$Be deposition modelled by SOCOL with that measured in Northern Finland, as shown in Figure 2. The agreement is very good: the average levels (183 and 188 Bq/m$^2$ for the modelled and measured data, respectively) agree within 2.5%, the correlation is highly significant ($r=0.86^{+0.04}_{-0.06}$, $p < 10^{-6}$). However, while the overall agreement is very good, the discrepancy can be significant for individual years. Thus, we conclude that the model reasonably well reproduces also the depositional flux of $^7$Be on the yearly time scale.

## 7  Conclusions

A full 3D transient model of production, transport, and deposition of cosmogenic $^7$Be and $^{10}$Be isotopes in the atmosphere has been developed. The model named as SOCOLv3.0:Be is based on the chemistry-climate model SOCOL, specifically tuned for the best performance in tracing beryllium, and the CRAC production model. Realistic modelling of $^7$Be isotope was performed for the years 2002–2008, with a 5-year model spin-up during 1996–2001. The measured weekly concentrations of $^7$Be in near-ground air have been compared with the model prediction for four nearly antipodal high-latitude locations, two in Northern (Finland and Canada) and two in Southern (Chile and Kerguelen Island) hemispheres. The model results generally agree well with the measurements in the absolute level within error bars, implying that the production, decay, and lateral deposition are correctly reproduced by the model. However, a larger discrepancy (up to 20%) was observed at the Kerguelen Island where the orographic scale is much smaller then the model grid size. The model correctly reproduces the temporal variability of $^7$Be concentrations on the annual and sub-annual scales, including a good reproduction of the annual cycle, which dominates data in the Northern hemisphere, and the absence of such a cycle in the Southern Hemisphere. In addition, occasional anomalous events such as SSWs are well-reproduced by the model. This validates the newly developed model to be able to correctly simulate the production, transport, and deposition of $^7$Be (and hence, $^{10}$Be) isotope on the local/regional spatial (in high-latitude regions) and monthly/annual temporal scales. The modelled $^7$Be distribution is also in general agreement with earlier computations based on a similar approach (e.g., Heikkilä et al., 2008a; Field et al., 2006).

We have also modelled the production and transport of $^7$Be for a major solar energetic-particle event of 20-Jan-2005, which was one of the largest directly observed events. It was shown that, to a minimum, an order of magnitude stronger event is needed to become observable in the beryllium data.

Concluding, a new full 3D time-dependent model, based on SOCOL-AERv2, of $^7$Be and $^{10}$Be atmospheric production, transport and deposition have been developed and validated using directly measured data. The model is recommended to be used in studies related to, e.g., atmospheric dynamical patterns, extreme solar particle storms, long-term solar activity reconstruction from cosmogenic proxy data, solar-terrestrial relation.

*Data availability.* The data from CTBTO are available free upon request at https://www.ctbto.org/specials/vdec/vdec-request-for-access. According to the EU's iNSPIRE directive, STUK provides measurement data connected to geospatial information free of charge. The data from STUK are available at https://www.stuk.fi/avoin-dataandhttps://www.stuk.fi/avoin-data/ohjeet. Data from the Radiation Protection Bureau of Health Canada are available free upon request via https://www.canada.ca/en/health-canada/corporate/contact-us/radiation-protection-bureau.html. SOCOL AER model is available at Zenodo https://doi.org/10.5281/zenodo.5006356. ERA5 data are available on the Copernicus Cli-

mate Change Service (C3S) Climate Data Store at https://cds.climate.copernicus.eu/#!/search?text=ERA5&type=dataset.

*Author contributions.* All authors participated in the model development, verification, discussions about the results, and revisions to the article. IU, GK and ER designed the main idea of this study and KG carried them out. KG was supervised directly by ER and TS during the model code improvement work. APL find and prepared the real Beryllium data for the model verification. IU and KG data processing and figures preparation. IU prepared the manuscript.

*Disclaimer.* The views expressed herein are those of the authors and do not necessarily reflect the views of the CTBTO Preparatory Commission.

*Acknowledgements.* The authors would like to thank Weihua Zhang, Ian Hoffman and Kurt Ungar of the Radiation Protection Bureau of Health Canada for providing the Yellowknife data. The authors acknowledge the work of CTBTO staff and CL18 and FRP30 station operating staff for producing and providing the corresponding data under a vDEC agreement (https://www.ctbto.org/specials/vdec/). This work was

partly supported by the Academy of Finland (Projects ESPERA no. 321882) and the Vilho, Yrjö and Kalle Väisälä Foundation of the Finnish Academy of Science and Letters (K. Golubenko has been granted a scholarship for 2021). The SOCOL model development and maintenance were supported by the Swiss National Science Foundation under grants 200021-169241 (VEC) and 200020-182239 (POLE). I. Usoskin work on the experimental design and parameterization of 7Be and 10Be production was partially supported by the Russian Science Foundation (RSF Project No. 20-67-46016). T. Sukhodolov work on the model development is a part of the SPbSU "Ozone Layer and Upper Atmosphere

Research laboratory" activity supported by the Government of the Russian Federation (075-15-2021-583).

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
