# Peer review of "Application of Chemistry-climate model SOCOL-AERv2-BEv1 to cosmogenic beryllium isotopes: Description and validation for polar regions"

_Geoscientific Model Development, 2021_

## Referee Comment (RC1)

**Revision of "Chemistry-climate model SOCOL-AERv2-BEv1 with the cosmogenic Beryllium-7 isotope cycle" by Golubenko K. et al.**

MS No: gmd-2021-56

MS Type: Development and Technical paper

*General comments*

This work presents a chemistry-climate model to simulate beryllium isotopes in the atmosphere. The capability of the model to simulate such beryllium isotopes concentrations is proven through a direct comparison of the simulations with near-ground observations at four stations. While the manuscript represents a valuable contribution to modeling science within the scope of Geoscientific Model Development, the presentation of the findings suffers from a number of significant flaws, for which reason it cannot be accepted for publication in its present state, but needs thorough substantial revisions.

First of all, the authors talk in and there of beryllium isotopes, but when it comes to the presentation of the results, one can find only results for beryllium-7 (which is also in the title), which generates a big confusion in the reader. This is not a major technical flaw, but truly does not help the reader to follow the work.

But now let's move with more significant revisions. The Introduction section does not convey successfully the need of this work and in particular of such a model for beryllium isotopes in the atmosphere. I mean, I cannot find there any technical information on the accuracy of previous models, or on the easy-to-use of such models, so there is no way for the reader to compare the model presented here with previous ones, and conclude about its improvement in one or more directions. I would suggest including more details on how previous models present significant gaps (with some data, if possible) so that the reader can understand immediately how this work goes in the direction of filling those gaps.

Moving to the validation of the model comparing the simulated values with observations at four stations in both hemispheres, which to me should be presented before the results of a particular event such as the SEP, I can see some limitations in the discussion and in the presentation of the accuracy, which is far from being fully validated as stated in the abstract and in and there in the manuscript. Indeed, the plots comparing simulated and observed values highlight that the model is not fully capable to catch the interannual variability, and especially in the southern hemispheres does not describe the observed pattern. The analysis of linear correlation coefficients and their significance is limited in this sense, since it does not provide information on the presence of biases but only on the similarity of the reproduced patterns. Additional statistical parameters would be needed to correctly conclude about the presence of biases. In addition, the discussion of the correlation coefficients for

stations located in the southern hemisphere is affected by significant flaws. Indeed, the low correlation coefficient found at these stations does not derive from the absence of a seasonal pattern, but instead highlights that the model is completely uncapable to correctly describe the variability of near-ground concentrations at those stations. The reasons of these disagreements, which may actually depend on a number of physical factors including an incorrect reproduction of deposition or transport, are not sufficiently investigated.

Also, the authors did not present anything of the meteorological data used in this chemistry-climate model, on which some of the disagreements between simulations and observations may actually depend. Indeed, even though the authors state that a gaseous deposition was adopted for beryllium isotopes, which is not sufficiently explained given that in reality beryllium isotopes travel attached to fine-sized aerosols and is thus mainly removed by wet deposition, they have searched for aerosol data when it came to explain some biases.

The discussion of the SEP event is far from being reasonable and well-given. Indeed, the fact that near-ground concentrations remain quite low, and that high beryllium concentrations increase only in the upper atmospheric layers, probably result from the particular meteorology of the period, which probably did not favor the transport of such high concentrations to the lower tropospheric layers. In addition, the discussion of the dependence on transport is achieved only by a shift of the date of the event to a different season, without giving additional details about the particular synoptic situation characterizing those days, which leaves the interpretation of the differences between the results mostly qualitative and somehow arbitrary.

To conclude, the authors present lots of technical details which pertain to the methods sections (e.g., information on measurement methods for beryllium, but also modeling information) in the results. I would suggest restructuring the paper to include those details in the methods so that the results section contains only the findings of this work and their appropriate discussion.

*Specific comments*

1) Title: I suggest including additional information in the title, such as: "Evaluation of…" or "Chemistry-climate …: description and evaluation", so that the title is more self-explanatory.
2) Line 1: is "probe" the most appropriate term? Wouldn't it be better to talk about "tracer"?
3) Lines 3-4: What do you mean by "such ready-to-use model"?
4) Lines 4-5 and following: Here you are talking about "isotopes of beryllium", but previously and in the title you were just referring to beryllium-7. Please check and modify as appropriate.
5) Lines 5-6: Which isotopes of beryllium?
6) Please use either $^7$Be either beryllium-7 all along the text to be consistent throughout the article.
7) Line 10: It is not clear which meteorological fields were used (from which model/reanalysis/…)

8) Lines 13-14: perhaps you could insert some statistical parameters proving your statements about the agreement of model simulations with the observations.

9) Line 16: it is not clear what you mean by "dominating data in the Northern Hemisphere".

10) Lines 26-27: Rephrase, this sentence is not clear.

11) Lines 34-36: The two sentences are quite obscure. Perhaps you could rephrase them as: "However, the transport of beryllium (isotopes?) in the atmosphere and its deposition on the surface or into the medium where it is measured may significantly affect the relationship between the production of the isotope and its content (or concentration) in the measured samples."

12) Lines 45-46: Could you provide some more details of the comparison of the model simulations with measurements (e.g., how many locations were compared?) and about the agreement between the model and the observations? This would provide the reader with indications and needs (or not) of the model presented in this paper.

13) Lines 47-48: Same as above, could you provide more details on this experiment and its results?

14) Lines 50-51: This sentence is quite obscure. In particular, it is not clear to me which feature is shared by the works of Pacini et al., 2011 and Brattich et al., 2020: indeed, while the first one presents an investigation of the depositional processes of $^7$Be-carrying aerosols in the troposphere using a combination of isotopic data with the numerical CRAC:Be7 model of cosmogenic production, the second one investigates the relationship between advection pathways and atmospheric composition (including natural radionuclides of terrestrial and cosmogenic origin) at a high mountain station, using back-trajectory cluster analysis. Possibly, the authors were referring to works like the ones of Liu H. et al. (e.g., Liu, Hongyu, Daniel J. Jacob, Isabelle Bey, and Robert M. Yantosca. 2001. "Constraints from 210Pb and 7Be on Wet Deposition and Transport in a Global Three-Dimensional Chemical Tracer Model Driven by Assimilated Meteorological Fields." Journal of Geophysical Research: Atmospheres 106 (D11) (June 16): 12109–12128. doi:10.1029/2000jd900839; Liu, H., Considine, D., Horowitz, W., Crawford, J., Rodriguez, S., Strahan, M., Damon, S., Steenrod, X., Xu, X., Kouatchou, J., Carouge, C., Yantosca, R. M., (2016). Using beryllium-7 to assess cross-tropopause transport in global models. Atmos. Chem. Phys. 16, 4641-4659, doi:10.5194/acp-16-4641-2016) or to other works from Brattich E. et al. (Brattich, E., Liu, H., Tositti, L., Considine, D. B., and Crawford, J. H.: Processes controlling the seasonal variations in $^{210}$Pb and $^7$Be at the Mt. Cimone WMO-GAW global station, Italy: a model analysis, Atmos. Chem. Phys., 17, 1061–1080, https://doi.org/10.5194/acp-17-1061-2017, 2017; Brattich, E., Liu, H., Zhang, B., Hernández-Ceballos, M. Á., Paatero, J., Sarvan, D., Djurdjevic, V., Tositti, L., and Ajtić, J.: Observation and modeling of high-$^7$Be events in Northern Europe associated with the instability of the Arctic polar vortex in early 2003, Atmos. Chem. Phys. Discuss. [preprint], https://doi.org/10.5194/acp-2020-1121, in review, 2021). If not, better clarifications of the links between the two cited papers from Pacini et al. and from Brattich et al. should be provided in the text.

15) Lines 51-52: I would assume that the knowledge of the wind field together with other meteorological parameters is a requisite for all dynamical atmospheric model, including the one presented here. Considering the temporal extension of current reanalysis (e.g., ERA5 climate reanalysis covering the period from 1950 on, or the NCEP/NCAR reanalysis from 1948 onwards), I cannot see how this actually limits the analysis of transport of beryllium isotopes, whose archives are not that longer.

16) Lines 59-69: This paragraph contains lot of technical data which are not fully pertinent to an Introduction section, while they should be moved to the Methods. Here you should provide a description of how the work presented here fills the gaps that you have just presented in the literature review above.

17) Lines 67-69: Referring to what I suggested at the previous point, I suppose that this should be moved to the Methods. In any case, information on the kind of observed meteorological data used in the model is missing, and should be also provided.

18) Lines 68-69: But why you purposedly chose to compare the model simulations just with measurement of $^7$Be (and not $^{10}$Be) at high-latitude stations? There are a lot of stations measuring cosmogenic isotopes located at midlatitudes or in the tropical/equatorial regions.

19) Lines 84-86: Is this information on the aerosol module connected with beryllium-isotopes? In the real atmosphere, there is a strong connection between 7Be and aerosol size distribution because it is known that after production 7Be rapidly attaches to submicron-sized particles, which is also very important for its removal through wet and dry deposition processes. This information is only partially provided in the text, with a partial description is given at lines 142. Better connection of these mechanisms with the way used in the model to simulate them should be provided.

20) Lines 92-93: There is no version number/year indicated, so there is no way for the reader to understand where the update is, the name is just the same as at line 89.

21) Line 108: Which isotope of beryllium?

22) Lines 109-110: To which time period are you referring to classify the strength of the event?

23) Lines 111-113: I understand that this information is from another recent paper by some of the authors, but I cannot clearly see how the shifting of the SEP event to another date can provide information on the seasonality of beryllium transport, if no information on the different transport or stability condition occurring during these dates is provided.

24) Lines 114-124: Any references for these sentences?

25) Lines 156-160: I am not convinced that the use of a gas deposition scheme instead than an aerosol deposition scheme is correct. Indeed, even though it is true that beryllium-isotopes attach to submicron-sized particles, it is well known that while precipitation scavenging is the dominant removal mechanism for aerosols (especially fine), the same is not totally true for gaseous species.

26) Lines 173-180: Rephrase, not clear.

27) Lines 181-188: Why are you using a multi-year mean? This way you are removing the interannual and seasonal variability. Wouldn't it better to investigate separately two seasons?

Any other studies showing similar observed/simulated patterns in deposition fluxes of $^7$Be to compare with?

28) Line 199: What do you mean by "caught by the air dynamics"?

29) Lines 203-208 and 209-212: Here you are describing the event from the point of view of model simulations, but is there any observations for you to document and compare your findings with? In this sense, Figure 5 shows that the $^7$Be concentration produced by the event were probably not transported at lower atmospheric layers, since probably notwithstanding the strength of the event, there was no transport of stratospheric-upper tropospheric air to the surface (at least in the model), which explains at least partially why the modelled activity of $^7$Be did not reach elevated values in near ground air in Finland.

30) Line 213: As described previously, this statement is not correct. Indeed, transport depends on many factors which can be seasonally dependent, but definitely depend on other characteristics that are not the seasons. If you do not provide information on the synoptic situation of the period, there is no way to conclude definitely that the transport of e.g., mid-autumn differs from the one of e.g., mid-spring.

31) Lines 213-223: But apart from the description of 7Be vertical cross sections in the different seasons, could you analyse the different transport mechanisms/synoptic situations occurring in the different seasons?

32) Lines 228-229: I understand that data availability is an important issue for any kind of model, but the use of weekly observations, which smooths lots of physical processes dominating the variability of beryllium-isotopes concentrations in the atmosphere poses great limitation to this comparison, which should be at least cited in the text.

33) Lines 228-233: Any references for the description of these measurements?

34) Lines 231-232: Please provide some additional details on the procedure to perform "standard correction for decay".

35) Line 235: Why do you use a set of stations for Finland while you use just one measuring station for the other locations. Could this then lead to a different result of the comparison? Explain.

36) Lines 228-264: Here the text comprehends also description of the measurement methods, which should be provided in a different (previous) section than this one.

37) Lines 247-248: As reported previously, the weekly information actually smooths the original signal, so I believe it could be important for the authors to show whether the model is able to catch the daily pattern of observations or not.

38) Lines 267-268 and below: The significance level and the value of the correlation coefficient provide information on the temporal coherence between the observed and simulated beryllium patterns, but does not provide information on the presence of a bias between observations and simulations. Below you provide a comparison between overall simulated and observed mean, but again this does not provide a true measure of bias. Please consider the inclusion of additional parameters for the comparison.

39) Line 273: Considering that the paper from Brattich et al. (2020) focuses on a mid-latitude high-altitude station, I doubt that there is any references in this paper with SSW events.

40) Lines 274-276 and below: Could you include a comparison of simulated and observed standard deviations?

41) Lines 277-279: Could you explain better why you suppose that such discrepancies between model and observations relate with atmospheric aerosol properties, especially since you described previously that you applied a gaseous deposition scheme? Couldn't the difference be related with a problem in the meteorological field (wind, precipitation, …)? In any case, what do you mean by "anomalies" in the "atmospheric aerosol properties"?

42) Lines 287-288 and 293: The absence of a seasonal pattern in the observed time series is not a justification of the absence of correlation between measurements and simulations (also, please note that the significance of the correlation, like of any other statistical parameters, is provided by the p-value, and not by the value of the coefficient) by itself, while it suggests that the model does not reproduce correctly the observed time pattern at these two stations, contradicting the statement in the text.

43) You talk about "orography" but how high is the sampling site? Did you compare model topography with real data?

44) Lines 298-299: Based on the reasonings provided above, it seems that the model reproduces correctly the patterns in the northern hemisphere, while the capability is more limited in the southern hemisphere, which may be related with additional factors than the orography and the spatial resolution (which should apply to all stations but in Finland where a set of different stations was used.

45) Figure 8: The figure shows very clearly how the model is able to catch the overall pattern, but is affected by some biases in reproducing some episodes, like for instance: a consistent overestimation for 2008 in Finland; a consistent underestimation of 2007 data in Finland; a period of underestimation in 2008 in Canada; general disagreement of the patterns for Chile and Kerguelen data. All these disagreements are not totally caught by the statistical parameters presented in the discussion, but need thorough investigation and discussion.

46) Figure 9: Also deposition data show similar disagreement not caught by the presented statistical parameters, and are not investigated.

47) Figure 10: The comparison of wavelet coherence between modeled and observed data is not sufficiently explained in the text, therefore the reader cannot properly understand the meaning of the panels.

48) Lines 309-310: You have presented results only for 7Be, so I am wondering you can claim the validity for all cosmogenic beryllium isotopes.

49) Line 315: I cannot see any error bars in the figures about comparison of the model with the observations.

50) Lines 316-317: Again, you talk about orography but there is no description of the model representation of the topography.

51) Lines 317-319: Based on my comments above, this sentence needs thorough revision.

52) Lines 324-325: Again, I suppose that the fact that you are not able to observe this event at near-ground is related more with the absence of transport from the stratosphere-upper troposphere than with the strength of the event.

53) Code and data availability: from the statement it is not clear that the observations used in this work are not freely available. Indeed, the website at STUK present a service price list, which probably means that the reader has to pay if wants to obtain the data, while data from the CTBTO seem to be available upon request (I did not proceed, so I cannot confirm that they are available for free upon request). To me, this seems in contrast with that the Code and data policy of the GMD journal (available at: https://www.geoscientific-model-development.net/policies/code_and_data_policy.html), which explicitly states that the data and other information underpinning the research findings are "findable, accessible, interoperable, and reusable" (FAIR). Regarding the licence of the model, it is not clear whether they are conform to the Open Source Definition. In addition, the document also states that: "Where the authors cannot, for reasons beyond their control, publicly archive part or all of the code and data associated with a paper, they must clearly state the restrictions. They must also provide confidential access to the code and data for the editor and reviewers in order to enable peer review. The arrangements for this access must not compromise the anonymity of the reviewers. All manuscripts which do not make code and data available at this level are to be rejected. Where only part of the code or data is subject to these restrictions, the remaining code and/or data must still be publicly archived. In particular, authors must make every endeavour to publish any code whose development is described in the manuscript. Code and data access must be provided at the time that the discussion paper is submitted. Embargoes, whether pending acceptance or for a defined period, are not acceptable.

And more:

1. the source code for the complete model or module or other coded product described in the paper (must be provided for model description, development and technical, and methods for assessment paper types);

2. the manual and any other model documentation (applies to model description, development and technical, and methods for assessment, to the extent the editor considers applicable);

3. all configuration files, boundary conditions, and input data (must be provided for experiment description papers and any other papers in which results from model runs are reported);

4. data sets for forcing of models or comparison with model output (must be provided for papers describing such data sets or for papers in which model output are compared with such data);

5. preprocessing, run control and postprocessing scripts covering every data processing action for all the results reported in the paper (applies for all papers, to the extent the editor considers applicable)."

So, overall, it seems to me that the statements of this section are not compliant with the requirements from this journal.

*Technical corrections*

1) Line 11: Change "the measured" with "observations".
2) Line 12: Change "cadence" with "time resolution" and "ones in" with "at".
3) Line 13: Change "hemispheres" to "hemisphere".
4) Line 19: Change "real data" with "observations".
5) Line 32, 37, 127: Change "long-living" with "long-lived".
6) Line 37: Change "probe" to "analyse".
7) Line 44: Replace "Model" with "model".
8) Line 61: I suppose that you did not use all "the available measurements". So I would recommend to change this to "available observations from different stations around the globe in both hemispheres." Possibly also add the time period of the measurements.
9) Line 63: Delete one "the".
10) Lines 105-108: I would suggest rephrasing: "While GCR are always present near the Earth, sporadic solar particle events (SEPs), which can be sufficiently strong to … and to produce a large amount of cosmogenic isotopes (…), take place occasionally (…)"
11) Line 109: Change "studied" to "study". Add "one" after "strongest".
12) Line 110: Here and throughout the manuscript: check the acronym, is it SEP or SPE?
13) Figure 1, caption: Change "model's" to "model". The units of the color scale is the same for both panels so there is no need to repeat the information for both panels and could be provided just in the general description of the figure before or after the description of the two panels.
14) Line 132: Change "the" to "a".
15) Line 158: Change "utilize" to "utilizes".
16) Line 170: Change "Boreal" to "boreal".
17) Line 172: Change "hemispheres" to "hemisphere".
18) Figure 3, caption: You should describe also the vertical information which is provided in the Figure.
19) Line 209: Add "one" before "strongest".
20) Figure 6: Explain the unit in the caption.
21) Figure 6, caption: The caption is probably not describing the figure correctly, since if the x-axis depicts days from 0 to 120 in 2005, it is not possible that the plot reports just the modelled activity of 20-Jan-2005 as reported in the caption.
22) Figure 7: Explain the unit in the caption and use a consistent unit for atoms (either "atom" either "at").
23) Line 247: Change "cadence" to "time resolution":
24) Line 258: Change "CTBT" to "CTBTO".
25) Line 259: The information provided between parenthesis is redundant since it is already contained in the fact that you use "total-deposition". Please delete.
26) Line 289: Change "suggests" with "suggesting".
27) Invert Figure 9 with Figure 10, as deposition is the last commented.

28) Lines 337-340: Revise use of tense in this section (all the same).
29) Line 338: Delete one "the" and change "them" to "it".
30) Line 339: Delete "real".
31) Lines 339-340: A verb is missing in this sentence.

---

## Author Comment (AC1)

We are grateful to the Reviewer for her/his thorough reading and detailed comments that led to an improvement of the manuscript, making it clearer. All the comments are addressed below and highlighted in the revised paper.

*"General comments*

*First of all, the authors talk in and there of beryllium isotopes, but when it comes to the presentation of the results, one can find only results for beryllium-7 (which is also in the title), which generates a big confusion in the reader. This is not a major technical flaw, but truly does not help the reader to follow the work. But now let's move with more significant revisions. The Introduction section does not convey successfully the need of this work and in particular of such a model for beryllium isotopes in the atmosphere. I mean, I cannot find there any technical information on the accuracy of previous models, or on the easy-to-use of such models, so there is no way for the reader to compare the model presented here with previous ones, and conclude about its improvement in one or more directions. I would suggest including more details on how previous models present significant gaps (with some data, if possible) so that the reader can understand immediately how this work goes in the direction of filling those gaps."*

We agree with this comment. We indeed are primarily focused on modelling of $^7$Be isotope as it provides a direct test for the beryllium production+transport+deposition model. On the other hand, in the future, we aim at full modelling of $^{10}$Be isotope which is produced and transported similarly to $^7$Be with only different decay time. That is needed for reconstructions of long-term solar variability and extreme SEP events. We are not aware of any directly validated full production+transport+deposition model of beryllium isotopes, that can be readily applicable to an analysis of past records, viz. without known meteorological data fields. We report such a model here, where the validation is performed vs. $^7$Be data. Since our primary goal is a validation of the beryllium model with the eventual application for $^{10}$Be data, we are focused mostly on high-latitude regions and annual time scales. We have *revised the Introduction* accordingly and added *Section 2 "Summary previous and existing models"*. We hope it is clearer for a reader now.

*"Moving to the validation of the model comparing the simulated values with observations at four stations in both hemispheres, which to me should be presented before the results of a particular event such as the SEP, I can see some limitations in the discussion and in the presentation of the accuracy, which is far from being fully validated as stated in the abstract and in and there in the manuscript. Indeed, the plots comparing simulated and observed values highlight that the model is not fully capable to catch the interannual variability, and especially in the southern hemispheres does not describe the observed pattern. The analysis of linear correlation coefficients and their significance is limited in this sense, since it does not provide information on the presence of biases but only on the similarity of the reproduced patterns. Additional statistical parameters would be needed to correctly conclude about the presence of biases. In addition, the discussion of the correlation coefficients for stations located in the southern hemisphere is affected by significant flaws. Indeed, the low correlation coefficient found at these stations does not derive from the absence of a seasonal pattern, but instead highlights that the model is completely uncapable to correctly describe the variability of near-ground concentrations at those stations. The reasons of these disagreements, which may actually depend on a number of physical factors including an incorrect reproduction of deposition or transport, are not sufficiently investigated."*

We agree that neither cross-correlation nor wavelet coherence can provide information about possible biases, and we have added a new plot (*see Fig. 11*) showing the distribution of the residual difference between the modelled and measured $^7$Be data. One can see that the null hypothesis of no

bias (the mean difference is indistinguishable from zero) cannot be rejected at any significance level, implying no bias even for the southern-hemisphere stations. Thus, both the time variability (coherence) and the absolute levels (no bias) of the data are reproduced by the model.

Also, the authors did not present anything of the meteorological data used in this chemistry-climate model, on which some of the disagreements between simulations and observations may actually depend. Indeed, even though the authors state that a gaseous deposition was adopted for beryllium isotopes, which is not sufficiently explained given that in reality beryllium isotopes travel attached to fine-sized aerosols and is thus mainly removed by wet deposition, they have searched for aerosol data when it came to explain some biases.

CCM SOCOL uses ECHAM5 (*see lines 174-175*) nudged with ERA5 (eraiaT42L39) reanalyses. ERA5 is the fifth generation ECMWF atmospheric reanalysis data of the global climate covering the period from January 1950 to the present. ERA5 data are available on the Copernicus Climate Change Service (C3S) Climate Data Store: https://cds.climate.copernicus.eu/#!/search?text=ERA5&type=dataset.

We improved the description of the transport as well as dry and wet deposition (*see lines 270-273, 275-278*). It can be found in the updated text and answers to the second reviewer.

The discussion of the SEP event is far from being reasonable and well-given. Indeed, the fact that near-ground concentrations remain quite low, and that high beryllium concentrations increase only in the upper atmospheric layers, probably result from the particular meteorology of the period, which probably did not favor the transport of such high concentrations to the lower tropospheric layers. In addition, the discussion of the dependence on transport is achieved only by a shift of the date of the event to a different season, without giving additional details about the particular synoptic situation characterizing those days, which leaves the interpretation of the differences between the results mostly qualitative and somehow arbitrary.

It is known that extreme SEP events in the past can be studied using [10]Be isotope in polar ice cores (e.g., Usoskin et al., 2006; Mekhaldi et al., 2015; Sukhodolov et al., 2018) that typically have the (pseudo)annual resolution. Here we wanted to check, both theoretically and experimentally, whether a weaker (strong but not extreme) SEP event can be observed in high-resolution beryllium data. As far as we know this question has not been fully addressed earlier. Moreover, as our modelling shows, the effect of a SEP event on the near-ground beryllium concentrations slightly depends on the season, because of the different patterns of the large-scale dynamics. During Summer-Autumn, the low tropopause and decreased static stability of the troposphere permit a more direct coupling with the upper atmosphere opening a path for the input of the polar stratospheric beryllium to lower levels. In contrast, in Winter-Spring, the tropopause rises, and intense radiative cooling stratifies the lower troposphere closing this route.

TTo conclude, the authors present lots of technical details which pertain to the methods sections (e.g., information on measurement methods for beryllium, but also modeling information) in the results. I would suggest restructuring the paper to include those details in the methods so that the results section contains only the findings of this work and their appropriate discussion.

We agree and have restructured the manuscript accordingly. Information on measurement methods for beryllium now in *Section 3*.

Specific comments

1) Title: I suggest including additional information in the title, such as: "Evaluation of…" or "Chemistry-climate …: description and evaluation", so that the title is more self-explanatory.

We included additional information in the title: "*Application of Chemistry-climate model SOCOL-AERv2-BEv1 to cosmogenic beryllium isotopes: Description and validation for polar regions.*"

2) Line 1: is "probe" the most appropriate term? Wouldn't it be better to talk about "tracer"?

It has been replaced with "tracer" (*see line 1*).

3) Lines 3-4: What do you mean by "such ready-to-use model"?

We mean that there were some earlier ad-hoc modelling efforts to demonstrate the ability of CCMs to model beryllium transport and deposition, but a user cannot use such models for other conditions because they are not readily available for a re-run. We cannot give full details in the abstract but explain them in the Introduction (*see lines 100-102*).

4) Lines 4-5 and following: Here you are talking about "isotopes of beryllium", but previously and in the title you were just referring to beryllium-7. Please check and modify as appropriate.

The model is applied to both $^7$Be and $^{10}$Be isotopes which are assumed to be transported and deposited similarly and only vary in decay. Thus, the model is indeed dealing with the isotopes of beryllium. However, since the high-precision measurements in air exist only for $^7$Be, we mostly discuss that. On the other hand, $^{10}$Be is discussed in the sense of SEP (*see Section 5.2*). We have clarified the text.

5) Lines 5-6: Which isotopes of beryllium?

Both $^7$Be and $^{10}$Be – see item 4 above.

6) Please use either 7Be either beryllium-7 all along the text to be consistent throughout the article.

Done, $^7$Be is used.

7) Line 10: It is not clear which meteorological fields were used (from which model/reanalysis/…)

CCM SOCOL uses ECHAM5 nudged with ERA5 (eraiaT42L39) reanalyses (*see lines 174-175*).

8)Lines 13-14: perhaps you could insert some statistical parameters proving your statements about the agreement of model simulations with the observations.

The model results agree with the measurements in the absolute level within error bars, implying that the production, decay and lateral deposition are correctly reproduced by the model (confidence level above 95%). We cannot provide more details in the abstract, but they are given in the main text (*see Fig. 10*).

9) Line 16: it is not clear what you mean by "dominating data in the Northern Hemisphere".

This means that the short-term variability of $^7$Be data in the N-hemisphere is dominated by the annual cycle (*see Figs. 1a and 1b*), while the annual cycle is hardly observed in the S-hemisphere. This pattern is well reproduced by the model.

10) Lines 26-27: Rephrase, this sentence is not clear.

The sentence has been changed (*see lines 34-35*).

11) Lines 34-36: The two sentences are quite obscure. Perhaps you could rephrase them as.

We have *revised the Introduction.*

12) Lines 45-46: Could you provide some more details of the comparison of the model simulations with measurements (e.g., how many locations were compared?) and about the agreement between the model and the observations? This would provide the reader with indications and needs (or not) of the model presented in this paper.

The text has been updated now (*see lines 72-77*).

13) Lines 47-48: Same as above, could you provide more details on this experiment and its results?

The text has been updated now (*see lines 81-90*).

14) Lines 50-51: This sentence is quite obscure. In particular, it is not clear to me which feature is shared by the works of Pacini et al., 2011 and Brattich et al., 2020: indeed, while the first one presents an investigation of the depositional processes of 7Be-carrying aerosols in the troposphere using a combination of isotopic data with the numerical CRAC:Be7 model of cosmogenic production, the second one investigates the relationship between advection pathways and atmospheric composition (including natural radionuclides of terrestrial and cosmogenic origin) at a high mountain station, using back-trajectory cluster analysis. Possibly, the authors were referring to works like the ones of Liu H. et al. (e.g., Liu, Hongyu, Daniel J. Jacob, Isabelle Bey, and Robert M. Yantosca. 2001. "Constraints from 210Pb and 7Be on Wet Deposition and Transport in a Global Three-Dimensional Chemical Tracer Model Driven by Assimilated Meteorological Fields." Journal of Geophysical Research: Atmospheres 106 (D11) (June 16): 12109–12128. doi:10.1029/2000jd900839; Liu, H., Considine, D., Horowitz, W., Crawford, J., Rodriguez, S., Strahan, M., Damon, S., Steenrod, X., Xu, X., Kouatchou, J., Carouge, C., Yantosca, R. M., (2016). Using beryllium-7 to assess cross-tropopause transport in global models. Atmos. Chem. Phys. 16, 4641-4659, doi:10.5194/acp-16-4641-2016) or to other works from Brattich E. et al. (Brattich, E., Liu, H., Tositti, L., Considine, D. B., and Crawford, J. H.: Processes controlling the seasonal variations in 210Pb and 7Be at the Mt. Cimone WMO-GAW global station, Italy: a model analysis, Atmos. Chem. Phys., 17, 1061–1080, https://doi.org/10.5194/acp-17-1061-2017, 2017; Brattich, E., Liu, H., Zhang, B., Hernández-Ceballos, M. Á., Paatero, J., Sarvan, D., Djurdjevic, V., Tositti, L., and Ajtić, J.: Observation and modeling of high7Be events in Northern Europe associated with the instability of the Arctic polar vortex in early 2003, Atmos. Chem. Phys. Discuss. [preprint], https://doi.org/10.5194/acp-2020-1121, in review, 2021). If not, better clarifications of the links between the two cited papers from Pacini et al. and from Brattich et al. should be provided in the text.

We mean that the discussed models are based on either a simplified 1D box-transport or back-tracing trajectory codes, without a full atmospheric dynamic modelling. The trajectory tracing approach requires the wind field to be accurately known and is typically applied to case studies. We have extended the discussion and added more references (see *lines 93-96*).

Lines 51-52: I would assume that the knowledge of the wind field together with other meteorological parameters is a requisite for all dynamical atmospheric model, including the one presented here. Considering the temporal extension of current reanalysis (e.g., ERA5 climate reanalysis covering the period from 1950 on, or the NCEP/NCAR reanalysis from 1948 onwards), I cannot see how this actually limits the analysis of transport of beryllium isotopes, whose archives are not that longer.

Potentially, a full CCM model can be run in a self-consistent mode (without nudging to the meteorological data) and still reasonably reproduce the beryllium transport+deposition, as demonstrated by Heikkilä et al. and Sukhodolov et al. This is our forthcoming step. But first, we need to demonstrate that the model works well with the nudging and then proceed to the self-consistent mode.

Lines 59-69: This paragraph contains lot of technical data which are not fully pertinent to an Introduction section, while they should be moved to the Methods. Here you should provide a description of how the work presented here fills the gaps that you have just presented in the literature review above.

This paragraph was rewritten to emphasize the aims of the paper. However, we cannot present the results of this work in the Introduction. It is done in the subsequent sections.

17) Lines 67-69: Referring to what I suggested at the previous point, I suppose that this should be moved to the Methods. In any case, information on the kind of observed meteorological data used in the model is missing, and should be also provided.

The text has been modified (*see lines 122-124*).

18) Lines 68-69: But why you purposedly chose to compare the model simulations just with measurement of 7Be (and not 10Be) at high-latitude stations? There are a lot of stations measuring cosmogenic isotopes located at midlatitudes or in the tropical/equatorial regions.

The main purpose of this work is to confront the model results with high-resolution (weekly) measurements of beryllium. $^7$Be was chosen for two reasons: first, there is almost no high-quality $^{10}$Be data in air samples since measurements of $^{10}$Be are much more difficult than those of $^7$Be; second, $^{10}$Be measured in ice cores contains an additional component (deposition) which is also not easy to model. On the other hand, the model eventually aims at the modelling of the $^{10}$Be isotope including a deposition. Since $^{10}$Be is typically measured at polar ice cores, we are primarily interested to validate the model for high-latitude regions. The text has been updated accordingly.

19) Lines 84-86: Is this information on the aerosol module connected with beryllium-isotopes? In the real atmosphere, there is a strong connection between 7Be and aerosol size distribution because it is known that after production 7Be rapidly attaches to submicron-sized particles, which is also very important for its removal through wet and dry deposition processes. This information is only partially provided in the text, with a partial description is given at lines 142. Better connection of these mechanisms with the way used in the model to simulate them should be provided.

The aerosol module is crucial for the proper modelling of beryllium transport, and thus, its brief description is needed for the benefit of a reader.

20) Lines 92-93: There is no version number/year indicated, so there is no way for the reader to understand where the update is, the name is just the same as at line 89.

The CRAC model does not contain the version number, it indicates the family of the model, viz. the way to model the atmospheric cascade. Each new release substitutes the earlier versions. The reference gives the unique identification of the model version.

21) Line 108: Which isotope of beryllium?

Both $^7$Be and $^{10}$Be were modelled.

22) Lines 109-110: To which time period are you referring to classify the strength of the event?

We refer to the period 1996-2008 as the "studied" period, and 1951—2020 for the "directly observed" events. The text has been clarified (*see lines 210-211*).

23) Lines 111-113: I understand that this information is from another recent paper by some of the authors, but I cannot clearly see how the shifting of the SEP event to another date can provide

information on the seasonality of beryllium transport, if no information on the different transport or stability condition occurring during these dates is provided.

We used the same SPE event with the same energy spectrum, viz. the same beryllium production pattern, for different seasons, when the actual atmospheric conditions were used. Accordingly, all the differences between panels in Figure 9 are caused by the different patterns of the atmospheric dynamics during different seasons. This forms an additional uncertainty in the relation between the SEP event strength and the expected beryllium signal. This uncertainty needs to be modelled and understood (see also Sukhodolov et al., 2017).

24) Lines 114-124: Any references for these sentences?

This comment was probably caused by our unclear writing. We now specify that this discussion is about the model results (not the measured features) shown in Figure 3. The text has been clarified (*see lines 215-218*).

25) Lines 156-160: I am not convinced that the use of a gas deposition scheme instead than an aerosol deposition scheme is correct. Indeed, even though it is true that beryllium-isotopes attach to submicron-sized particles, it is well known that while precipitation scavenging is the dominant removal mechanism for aerosols (especially fine), the same is not totally true for gaseous species.

The wet-deposition scheme uses uptake of the species by liquid water in-cloud and rain droplets below the clouds which depend on the solubility and reactivity of the component. For our study, these parameters for the beryllium isotopes were set equal to those for fine sulfate-containing aerosols. We believe that a more precise description of the wet beryllium-isotope deposition is not straightforward, because it is not clear what kind of aerosol (there are many different aerosol types in the troposphere) or ion clusters is more attractive for Be isotopes to attach to, and how to treat cases when aerosol is fully evaporated? Our approach is obviously a simplification, but the obtained results confirm the applicability of the applied method. We work on a better scheme to estimate, as our aerosol model treats interactively only sulfur-containing aerosols that is probably not sufficient for the troposphere.

26) Lines 173-180: Rephrase, not clear.

*See lines 290-298*.

27) Lines 181-188: Why are you using a multi-year mean? This way you are removing the interannual and seasonal variability. Wouldn't it better to investigate separately two seasons? Any other studies showing similar observed/simulated patterns in deposition fluxes of 7Be to compare with?

This is only an illustration to show that depositions can vary by an order of magnitude for different locations.

28) Line 199: What do you mean by "caught by the air dynamics"?

The sentence is modified (*see line 315*).

29) Lines 203-208 and 209-212: Here you are describing the event from the point of view of model simulations, but is there any observations for you to document and compare your findings with? In this sense, Figure 5 shows that the 7Be concentration produced by the event were probably not transported at lower atmospheric layers, since probably notwithstanding the strength of the event, there was no transport of stratospheric-upper tropospheric air to the surface (at least in the model), which explains at least partially why the modelled activity of 7Be did not reach elevated values in near ground air in Finland.

The data suggest that there is no clear signal of $^7$Be in measurements (see also Usoskin et al., 2009), as the expected SEP-related $^7$Be activity is a factor 100 lower than that from GCR. This cannot be directly checked with measurements, since SEP- and GCR-produced beryllium cannot be distinguished in data. On the other hand, we know that extreme SEP events (e.g., in 775 AD, 993 AD, 660 BC) can be clearly observed in polar $^{10}$Be (Mekhaldi et al., 2015; Sukhodolov et al., 2017: O'Hare et al., 2019). Thus, it is worth modelling this.

30) Line 213: As described previously, this statement is not correct. Indeed, transport depends on many factors which can be seasonally dependent, but definitely depend on other characteristics that are not the seasons. If you do not provide information on the synoptic situation of the period, there is no way to conclude definitely that the transport of e.g., midautumn differs from the one of e.g., mid-spring.

The transport of beryllium after production depends on many different factors such as QBO phase, the appearance of sudden stratospheric warmings or perturbations of the wave patterns, but the seasonal behaviour of the transport usually dominates.

31) Lines 213-223: But apart from the description of 7Be vertical cross sections in the different seasons, could you analyse the different transport mechanisms/synoptic situations occurring in the different seasons?

The main features of the transport during cold and warm seasons are well known. Synoptic situations are usually very short and sporadic considered by the model via the nudging. We do not see how they can help to understand seasonal effects.

32) Lines 228-229: I understand that data availability is an important issue for any kind of model, but the use of weekly observations, which smooths lots of physical processes dominating the variability of beryllium-isotopes concentrations in the atmosphere poses great limitation to this comparison, which should be at least cited in the text.

The problem with daily-resolved data is that it is very noisy partly due to local weather conditions e.g., local rain showers, and this is not generally captured by a model with a grid size of 2.5X2.5° The use of larger sampling frequency tend to average out local weather events. We found the weekly-resolution data optimal for our purposes. The text has been updated (*see lines 129-135*).

33) Lines 228-233: Any references for the description of these measurements?

This article has been cited already as Leppänen et al., 2012 (*see line 138*).

34) Lines 231-232: Please provide some additional details on the procedure to perform "standard correction for decay".

We apologize for the unclear writing. Here we refer to the standard procedure of the correction for decay which occurs between the sample collection and measurements. This is a standard correction applied to all datasets before their release and is not a part of our model (see a detailed description of the sampling and measurement procedure, e.g., in Leppänen et al., 2012 or STUK and CTBTO websites). For our model, "correction for decay" was done in section 3.2 and implies correction for decay within the model, viz. between production and deposition.

35) Line 235: Why do you use a set of stations for Finland while you use just one measuring station for the other locations. Could this then lead to a different result of the comparison?

Finland is very well covered by measurement stations operated by STUK. Thanks to that, we can combine data to cover a region comparable to the model grid cell thus avoiding very local effects. As a comparison with data shows (Figs. 1a, 2, 10a), the agreement is good. For other locations, we do not have data from such a dense regional network, since CTBTO provides data from a much sparser grid. The effect of the mismatch between the model grid and the local scale is particularly important for Kerguelen where the size of the island is significantly smaller than the model cell. Thus, we can study the effect of the grid size.

36) Lines 228-264: Here the text comprehends also description of the measurement methods, which should be provided in a different (previous) section than this one.

The text has been restructured as proposed by the Reviewer.

37) Lines 247-248: As reported previously, the weekly information actually smooths the original signal, so I believe it could be important for the authors to show whether the model is able to catch the daily pattern of observations or not.

See our reply to items 12 and 32 above.

38) Lines 267-268 and below: The significance level and the value of the correlation coefficient provide information on the temporal coherence between the observed and simulated beryllium patterns, but does not provide information on the presence of a bias between observations and simulations. Below you provide a comparison between overall simulated and observed mean, but again this does not provide a true measure of bias. Please consider the inclusion of additional parameters for the comparison.

A new Figure 11 with the distribution of residuals is included (see also our reply to general comments above). The analysis shows that the null hypothesis of no bias cannot be rejected at any sensible statistical level, implying effectively that the model does not produce any significant bias.

39) Line 273: Considering that the paper from Brattich et al. (2020) focuses on a mid-latitude high-altitude station, I doubt that there is any references in this paper with SSW events.

We apologize for the typo with the wrong reference. The correct reference is Brattich et al., 2021, where the effect of SSW is discussed.

40) Lines 274-276 and below: Could you include a comparison of simulated and observed standard deviations?

The standard deviations (in units of mBq/m3) of the simulated/observed weekly data are: 0.97/1.04 for Finland, 1.78/1.87 for Canada, 0.35/0.5 for Chile, and 0.38/0.48 for Kerguelen. The standard deviations are, of course, larger for Finnish and Canadian locations where the variability is dominated by the seasonal cycle, and smaller for Chilean and Kerguelen locations where it represents the synoptic noise. In all cases, the difference is 0.1-0.15 mBq/m3. A clearer picture is shown in the new *Figure 11*.

Lines 277-279: Could you explain better why you suppose that such discrepancies between model and observations relate with atmospheric aerosol properties, especially since you described previously that you applied a gaseous deposition scheme? Couldn't the difference be related with a problem in the meteorological field (wind, precipitation, …)? In any case, what do you mean by "anomalies" in the "atmospheric aerosol properties"?

The applied deposition scheme is based on the sulfate aerosol properties, but if [7]Be is attached to other kinds of particles presenting in the troposphere the deposition and transport can be rather different leading to some inaccuracy of the simulations. Of course, the problem with the transport modelling and meteorological fields are also important.

The word "anomalies" in the "atmospheric aerosol properties" is related to the scale of aerosol optical depth which is a measure of the extinction of the solar beam by the dust and haze. So, particles in the atmosphere (dust, smoke, pollution, volcanic activity) can block sunlight by absorbing or by scattering the light.

42) Lines 287-288 and 293: The absence of a seasonal pattern in the observed time series is not a justification of the absence of correlation between measurements and simulations (also, please note that the significance of the correlation, like of any other statistical parameters, is provided by the p-value, and not by the value of the coefficient) by itself, while it suggests that the model does not reproduce correctly the observed time pattern at these two stations, contradicting the statement in the text

We always provide the p-value for correlation coefficients. The signal at each site is composed of the annual cycle and the short-scale synoptic variability. The model correctly reproduces the presence (or absence) of the annual cycle but can deviate from the data for the high-frequency noise. Thus, the correlation is high and significant for sites with the dominant annual cycle but low for sites without this cycle. However, the cross-correlation coefficient is confusing since it mixes up all frequency/time scales and mostly represents the dominant one (e.g., the annual cycle for the northern hemisphere). More representative is the coherence (Fig. 10), which localizes the "correlation" at different frequencies (time scales). It is obvious from the Figure that the time variability of [7]Be is correctly reproduced at the annual and sub-annual scale for all stations (coherence at 1-year scale is always highly significant being ~0.9).

43) You talk about "orography" but how high is the sampling site? Did you compare model topography with real data?

The model has 39 vertical levels between the Earth's surface and the 0.01 hPa level (≈80 km). We used the lowest model level (the Earth's surface). The model grid size is larger than the size of the Kerguelen island, thus the model cannot correctly catch their spatial scale.

44) Lines 298-299: Based on the reasonings provided above, it seems that the model reproduces correctly the patterns in the northern hemisphere, while the capability is more limited in the southern hemisphere, which may be related with additional factors than the orography and the spatial resolution (which should apply to all stations but in Finland where a set of different stations was used

We agree with this statement which correctly summarizes our results. On the other hand, we want to emphasize that the annual and sub-annual variability is always correctly reproduced (*see Fig. 10*) as well as the overall level, by the model.

45) Figure 8: The figure shows very clearly how the model is able to catch the overall pattern, but is affected by some biases in reproducing some episodes, like for instance: a consistent overestimation for 2008 in Finland; a consistent underestimation of 2007 data in Finland; a period of underestimation in 2008 in Canada; general disagreement of the patterns for Chile and Kerguelen data. All these disagreements are not totally caught by the statistical parameters presented in the discussion, but need thorough investigation and discussion

We agree that the model has some periods of not perfect reproduction of the data even for the Northern hemisphere, but statistically, the general pattern is very well caught (see Figures 10 and 11). Quantitative analysis (Fig.10) implies that the agreement is highly significant (at least at the time scales longer than 6 months) and has no essential biases (*Figure 11*). We recall that we are primarily interested in this annual/interannual time scale. The text has been clarified (*lines 369-371, 374-379*).

46) Figure 9: Also deposition data show similar disagreement not caught by the presented statistical parameters, and are not investigated.

The situation is similar to that discussed in item 45 above. Although each individual year is not perfectly reproduced, the main annual cycle is correctly reproduced in a statistical sense. In particular, the mean difference between modelled and measured values is only 4.7 Bq/m2 (~4%), as shown in the plot below.

[Figure]

47) Figure 10: The comparison of wavelet coherence between modeled and observed data is not sufficiently explained in the text, therefore the reader cannot properly understand the meaning of the panels.

The description of the wavelet coherence has been extended. *Lines 342-354*.

48) Lines 309-310: You have presented results only for 7Be, so I am wondering you can claim the validity for all cosmogenic beryllium isotopes.

Different isotopes of beryllium ($^7$Be and $^{10}$Be) are supposed to be transported in the atmosphere in similar ways (see, e.g., works by Heikkilä et al.). Accordingly, the results obtained for $^7$Be are suggestive also for $^{10}$Be. On the other hand, this is only a suggestion.

49) Line 315: I cannot see any error bars in the figures about comparison of the model with the observations.

We apologize for unclear writing confusing a reader. In this sentence we mean not the uncertainties of the measurements that are of the order of 7-8 %, but the standard deviation of the difference between modelled and measured activities (*Figure 11*).

50) Lines 316-317: Again, you talk about orography but there is no description of the model representation of the topography.

The real orography is smoothed for our grid cells (approximately 300x300 km). Therefore, local small-scale features cannot be well reproduced.

51) Lines 317-319: Based on my comments above, this sentence needs thorough revision.

The sentence has been reformulated to be more precise. *See lines 400-403*.

52) Lines 324-325: Again, I suppose that the fact that you are not able to observe this event at near-ground is related more with the absence of transport from the stratosphere-upper troposphere than with the strength of the event.

Both the lack of fast vertical transport, leading to decay of the stratospheric $^7$Be before it can reach the ground, and the strength of the event affects the detectability of a SEP-related signal in the near-ground air. As one can see from Fig. 8, an x100 stronger event would double the $^7$Be activity in Finland within 10-20 days, which would have been clearly measured.

53) Code and data availability: from the statement it is not clear that the observations used in this work are not freely available. Indeed, the website at STUK present a service price list, which probably means that the reader has to pay if wants to obtain the data, while data from the CTBTO seem to be available upon request (I did not proceed, so I cannot confirm that they are available for free upon request). To me, this seems in contrast with that the Code and data policy of the GMD journal (available at: https://www.geoscientific-modeldevelopment.net/policies/code_and_data_policy.html), which explicitly states that the data and other information underpinning the research findings are "findable, accessible, interoperable, and reusable" (FAIR). Regarding the licence of the model, it is not clear whether they are conform to the Open Source Definition. In addition, the document also states that: "Where the authors cannot, for reasons beyond their control, publicly archive part or all of the code and data associated with a paper, they must clearly state the restrictions. They must also provide confidential access to the code and data for the editor and reviewers in order to enable peer review. The arrangements for this access must not compromise the anonymity of the reviewers. All manuscripts which do not make code and data available at this level are to be rejected. Where only part of the code or data is subject to these restrictions, the remaining code and/or data must still be publicly archived. In particular, authors must make every endeavour to publish any code whose development is described in the manuscript. Code and data access must be provided at the time that the discussion paper is submitted. Embargoes, whether pending acceptance or for a defined period, are not acceptable.

We have clarified the data statement now. See *Data availability*.

1. the source code for the complete model or module or other coded product described in the paper (must be provided for model description, development and technical, and methods for assessment paper types);
2. the manual and any other model documentation (applies to model description, development and technical, and methods for assessment, to the extent the editor considers applicable);
3. all configuration files, boundary conditions, and input data (must be provided for experiment description papers and any other papers in which results from model runs are reported);
4. data sets for forcing of models or comparison with model output (must be provided for papers describing such data sets or for papers in which model output are compared with such data);
5. preprocessing, run control and postprocessing scripts covering every data processing action for all the results reported in the paper (applies for all papers, to the extent the editor considers applicable)."

All files are uploaded to Zenodo https://doi.org/10.5281/zenodo.5006356.

---

## Author Comment (AC2)

We are grateful to the Reviewer for her/his detailed comments. All the comments are addressed below and highlighted in the revised paper.

**Major comments:**

It appears that the authors are not fully aware of other existing global models that have been used to simulate atmospheric Be-7. For example, at the beginning of the abstract (or similarly on P2, L50-51), it is stated that "Previously, modelling of the beryllium atmospheric transport was performed using simplified box-models or air back-tracing codes. While the ability of full atmospheric dynamics models to model beryllium transport was demonstrated earlier, no such ready-to-use model is currently available." There has been a long history of simulating Be-7 using global models, e.g., Brost, R.A., J. Feichter, and M. Helmann, Three-dimensional simulation of 7Be in a global climate model, JGR, 96, 22,423-22,445, 1991 (https://agupubs.onlinelibrary.wiley.com/doi/abs/10.1029/91JD02283). The authors mentioned a few modeling papers (sometimes not accurately; see below for example) but there are many more.

We thank the reviewer for these suggestions. We have extended the discussion and added more references: *lines 66-90.*

P2, L47-48: "A full 3D modelling of the production and transport of beryllium isotopes in the Earth's atmosphere was performed earlier using the ECHAM5-HAM atmospheric model (Heikkilä et al., 2008a,b)." -- Heikkila et al. (2008a) used a two-box model and did not use a 3-D model. Heikkila et al. (2008b) used the production rates from Masarik and Beer (1999) and did not do a full 3-D modeling of the production of beryllium isotopes. P14, L273: Brattich et al. (2020) is not relevant to sudden stratospheric warming (SSW) events at all.

The sentence has been changed (*see line 81*).
We apologize for the typo with the wrong reference. The correct reference is Brattich et al., 2021, where the effect of SSW is discussed (*line 348*).

P3, L53-55: "while several models of different complexity and accuracy have been developed in the recent past to model transport and deposition of beryllium isotopes, most of them have been abandoned and not supported further and cannot be directly applied in new analysis works." – Which "several models"? Which ones were abandoned and not supported further? There are other existing global models (see point 1 above). Are

you saying that a global model of transport and deposition coupled with a Be-7 production model is needed? Was the CRAC:Be model coupled with SOCOL previously? More generally, it would help to list (in a table or schematic with references) the model components that already existed and those that this paper would like to develop or improve. The evaluation or performance of the original SOCOL model in simulating Be-10 also needs a bit of elaboration. While this paper focuses on Be-7, the same processes (except decay) control Be-7 and Be-10 in the troposphere.

We have changed this paragraph: *lines 103-123*.

Section 2.3: "beryllium is considered as a gas tracer" – This is confusing. As authors also stated, after production, Be-7 attaches to ambient aerosols. That's why Be-7 has long been used as an aerosol tracer. Therefore it should be treated as an aerosol in the model. It should not be treated as "gas form" as also stated in the last sentence of this paragraph.

Gas and aerosol forms of beryllium isotopes have similar transport for non-volcanic conditions due to the small size of particles. We have revised part 4.3: *see lines 246-256*.

Section 2.3: How is convective transport represented in the model? How about turbulent mixing in the boundary layer? How realistic is the stratosphere-to-troposphere transport of Be-7 (or other tracers)?

We have added to the text: *lines 260-264*.

Different processes such as stratospheric mixing, stratosphere–troposphere exchange, tropospheric transport and deposition, are realistically modelled by the CCM SOCOL (e.g., Feinberg et al., 2019).

Evaluation of the STT of Be-7 is one of this manuscript aims. Judging from the results it seems realistic (*see fig.5, 9*).

Section 2.4: "tropospheric washout of gases is calculated by…" - Be-7 is an aerosol tracer.

section 2.4: "Deposition of beryllium isotopes is parameterized as a function of surface properties, solubility and reactivity of the considered species (Kerkweg et al., 2006). This scheme considers actual meteorological conditions, different surface types, and trace gas properties like solubility and reactivity. Since beryllium is transported like a gas in

the CCM SOCOL, the dry deposition scheme is like other gases in the model (e.g., Revell et al., 2018). Moist convection contributes significantly to transport of energy, momentum, water, and trace gases in global modelling." - Again, Be-7 should be treated as aerosol (not gas) in both dry deposition and wet deposition parameterizations.

P8, L159-160: "Scavenging coefficients for gas-phase species are calculated based on Henry's law equilibrium constants." - If Be-7 is treated like a gas, it means Henry's law has been applied to Be-7 (actually an aerosol tracer) in the model, which does not make sense. Since scavenging is the largest Be-7 sink in the troposphere, more detailed description of the scavenging scheme is required here beyond simply citing the reference of Tost et al. (2010), for example, how large-scale (stratiform) vs. convective scavenging and in-cloud vs below-cloud scavenging are separately treated.

As we said before, gas and aerosol forms of beryllium isotopes have similar transport for non-volcanic conditions due to the small size of particles (e.g., Lal & Peters, 1967; Delaygue et al., 2015). However, for the dry- and wet-deposition schemes, we use Henry's constants and reactivity of the sulphate aerosol (*lines 279-280*).

We add more explanation to the text: *lines 275-278.*

A detailed description of the interactive wet-deposition scheme has been presented and discussed by Tost at al. (2006, 2007, 2010).

The parameterization is based on the model generated available liquid water in clouds (cloud water content) and below cloud (precipitating water) and uptake/release from droplets, which depends on the concentration and solubility of the considered species.

P12, L218: a) if SPE-produced Be-7 is hardly detectable in the background, please explain why it is still necessary or interesting to study the transport of SPE-produced beryllium; b) The reason for differences in seasonal transport is not given. Is it because of the seasonal minimum of stratosphere-to-troposphere transport in fall? L221: what's the faster removal mechanism in winter?

a) Because of the much softer energy spectrum, SEPs produce [7]Be at shallower atmospheric depths and higher latitudes than GCR do. The SPE-related 7Be signal is indeed unobservable for the recent decades, however, a factor of ~100 stronger SPEs are known to appear in the past (e.g., 775 AD. 994 AD) which have a clear signature in 10Be

records in polar ice cores. A proper model of beryllium transport/deposition is needed for an accurate analysis of such events. We have revised the Introduction to make it clearer.

b) As far as we know this question has not been fully addressed earlier. Moreover, as our modelling shows, the effect of a SEP event on the near-ground beryllium concentrations slightly depends on the season, because of the different patterns of the large-scale dynamics. During Summer-Autumn, the low tropopause and decreased static stability of the troposphere permit a more direct coupling with the upper atmosphere opening a path for the input of the polar stratospheric beryllium to lower levels. In contrast, in Winter-Spring, the tropopause rises, and intense radiative cooling stratifies the lower troposphere closing this route.

What's Be-7 residence time against deposition in this model, as compared to those in other models (e.g., Brost et al. 1991; Koch et al., 1996, https://agupubs.onlinelibrary.wiley.com/doi/abs/10.1029/96JD01176)? Simulated surface Be-7 concentrations are sensitive to wet/dry deposition.

The decrease of $^7$Be isotope concentration is nearly perfectly exponential with $t_{7Be} = 72\pm3$ days, which includes both decay and removal. Thus, the $^7$Be isotope is fully removed from the atmosphere, mostly due to decay, within several months.

**More comments:**

11) The text uses the word "beryllium isotopes" a lot but the paper mainly deals with Be-7 (and occasionally Be-10). Can you just say Be-7 (or Be-10)?

Done

12) Abstract: "An interactive deposition scheme was applied including both wet and dry depositions" - I don't think you applied a single deposition scheme that include both wet and dry deposition. L10: you actually presented results for 2002 (Fig.8), so it's not a spinup year. By "lateral deposition", do you mean surface deposition? "including a perfect reproduction of the annual cycle" – I don't think it's perfect (see Fig.8). Please avoid using the word "perfect" in the text.

Thanks for the corrections. See *Line 18 and line 401*.

13) "Comparison with the real data of 7Be concentration in the near-ground air fully validates the model and its high accuracy." – Comparison

with surface Be-7 observations from a limited number of locations does not fully validates the model. Again, I suggest the authors look up current literatures especially those on global modeling of Be-7, where information on global data sets of surface Be-7 concentrations, deposition fluxes, and/or high-altitude observations are available.

Eventually, we aim at the modelling of radionuclides deposition in Greenland and Antarctic ice sheets. Therefore we are focused mostly on high-latitude regions and annual time scales. The Introduction has been revised to explain this.

14) P2, L40-41 (also see P6, L126): "these models cannot be applied for the short-living 7Be isotope, whose half-life time is shorter than the typical atmospheric transport time" – Typical transport timescale in the troposphere is only ~hours to days.

We apologize for the typo. This sentence is revised.

15) P5, Figure 1: It is interesting to compare the production rates of Be-7 produced by GCR and SPE (even though they differ by magnitudes). However, these two panels use different units, making it hard to compare. Could you represent the production rates by SPE in "rates" instead of total production?

We have revised the plot and its description. Since the duration of a strong SPE is several hours, up to a day, we compare it with the daily production of beryllium by GCR. The use of production rate for an SPE makes little sense since it can vary by many orders of magnitude within short times.

16) Figure 9: specify in caption which two stations and their locations (latitude/longitude). Y-Title should be "Deposition (Bq/m2/TIME)" since each quarter may contain different hours.

We added to plot Bq/m2/3months and created new *Table 1* (List of stations whose data were used for the present study.).

17) P14, L277-279: Why and how?

Gas and aerosol forms of beryllium isotopes have similar transport for non-volcanic conditions due to the small size of particles (e.g., Lal & Peters, 1967; Delaygue et al., 2015). After a strong volcanic eruption, the size distribution can be shifted to larger values. So, if in this time period we have an essential difference between the model and the measurements

we may suggest that it happens due to the model using 7Be as a gas tracer. Using the AERONET we see  no aerosols anomalies for that time.

18) P15, L321-322: "The modelled beryllium distribution is also in general agreement with earlier computations based on a similar approach." --  Which earlier work?

We have added this sentence. *Line 406*.

---

## Referee Report (RR1)

**Revision of "Chemistry-climate model SOCOL-AERv2-BEv1 with the cosmogenic Beryllium-7 isotope cycle" by Golubenko K. et al.**

MS No: gmd-2021-56

MS Type: Development and Technical paper

*General comments*

The authors have replied to all my comments. However, in some cases the response is not completely satisfactory. Below I report my previous major comments in normal style, followed by their response highlighted in yellow, and followed by my new comment in italic.

First of all, the authors talk in and there of beryllium isotopes, but when it comes to the presentation of the results, one can find only results for beryllium-7 (which is also in the title), which generates a big confusion in the reader. This is not a major technical flaw, but truly does not help the reader to follow the work. But now let's move with more significant revisions. The Introduction section does not convey successfully the need of this work and in particular of such a model for beryllium isotopes in the atmosphere. I mean, I cannot find there any technical information on the accuracy of previous models, or on the easy-to-use of such models, so there is no way for the reader to compare the model presented here with previous ones, and conclude about its improvement in one or more directions. I would suggest including more details on how previous models present significant gaps (with some data, if possible) so that the reader can understand immediately how this work goes in the direction of filling those gaps.

We agree with this comment. We indeed are primarily focused on modelling of 7Be isotope as it provides a direct test for the beryllium production+transport+deposition model. On the other hand, in the future, we aim at full modelling of 10Be isotope which is produced and transported similarly to 7Be with only different decay time. That is needed for reconstructions of long-term solar variability and extreme SEP events. We are not aware of any directly validated full production+transport+deposition model of beryllium isotopes, that can be readily applicable to an analysis of past records, viz. without known meteorological data fields. We report such a model here, where the validation is performed vs. 7Be data. Since our primary goal is a validation of the beryllium model with the eventual application for 10Be data, we are focused mostly on high-latitude regions and annual time scales. We have *revised the Introduction* accordingly and added *Section 2 "Summary previous and existing models".* We hope it is clearer for a reader now.

*In this case I am rather satisfied with the revision. There is still confusion regarding the application of the model to beryllium isotopes and then its presentation and validation considering just Be-7.*

*Also, I do not completely understand the connection between a beryllium model with eventual application to Be-10 data with the choice of high-latitude regions and annual timescales. Could you please better specify this? As commented below in the specific comments, there are still some deficiencies in the added Section 2. Indeed, the review of earlier models is far from being complete, missing important works on this subject. It is true that this field is still widely open for improvements, but the state-of-the-art in this research field has to be consistently described. Secondly, to me the newly added Section is part of the Introduction section, being fundamental for the reader to understand the need of this particular work and its scope. Therefore, to me the introduction of this as Section 2 generates confusion and does not fully help to address the above reported issues.*

Moving to the validation of the model comparing the simulated values with observations at four stations in both hemispheres, which to me should be presented before the results of a particular event such as the SEP, I can see some limitations in the discussion and in the presentation of the accuracy, which is far from being fully validated as stated in the abstract and in and there in the manuscript. Indeed, the plots comparing simulated and observed values highlight that the model is not fully capable to catch the interannual variability, and especially in the southern hemispheres does not describe the observed pattern. The analysis of linear correlation coefficients and their significance is limited in this sense, since it does not provide information on the presence of biases but only on the similarity of the reproduced patterns. Additional statistical parameters would be needed to correctly conclude about the presence of biases. In addition, the discussion of the correlation coefficients for stations located in the southern hemisphere is affected by significant flaws. Indeed, the low correlation coefficient found at these stations does not derive from the absence of a seasonal pattern, but instead highlights that the model is completely uncapable to correctly describe the variability of near-ground concentrations at those stations. The reasons of these disagreements, which may actually depend on a number of physical factors including an incorrect reproduction of deposition or transport, are not sufficiently investigated.

We agree that neither cross-correlation nor wavelet coherence can provide information about possible biases, and we have added a new plot (*see Fig. 11*) showing the distribution of the residual difference between the modelled and measured 7Be data. One can see that the null hypothesis of no bias (the mean difference is indistinguishable from zero) cannot be rejected at any significance level, implying no bias even for the southern-hemisphere stations. Thus, both the time variability (coherence) and the absolute levels (no bias) of the data are reproduced by the model.

*As commented below, the histograms shown in Figure 11 actually indicate that apart from the mean value of the bias, there are many cases, especially in southern stations, when the bias is actually higher and close to 1. Indeed, the mean low value could result from the distribution of the bias. In addition, the histogram may not be the best way to represent the bias, since the shape of the histogram is dominated by the choice of the bin width. As I previously mentioned, there are a set of statistical parameters to calculate and to validate a model. In addition, it would be interesting to evaluate the pattern of the bias to investigate if it is higher during particular seasons or if conversely the high biases are randomly distributed.*

Also, the authors did not present anything of the meteorological data used in this chemistry-climate model, on which some of the disagreements between simulations and observations may actually depend. Indeed, even though the authors state that a gaseous deposition was adopted for beryllium isotopes, which is not sufficiently explained given that in reality beryllium isotopes travel attached to fine-sized aerosols and is thus mainly removed by wet deposition, they have searched for aerosol data when it came to explain some biases.

CCM SOCOL uses ECHAM5 (*see lines 174-175*) nudged with ERA5 (eraiaT42L39) reanalyses. ERA5 is the fifth generation ECMWF atmospheric reanalysis data of the global climate covering the period from January 1950 to the present. ERA5 data are available on the Copernicus Climate Change Service (C3S) Climate Data Store: https://cds.climate.copernicus.eu/#!/search?text=ERA5&type=dataset. We improved the description of the transport as well as dry and wet deposition (*see lines 270-273, 275-278*). It can be found in the updated text and answers to the second reviewer.

*Ok with this addition. However, it is still not clear which kind of meteorological data have been used.*

The discussion of the SEP event is far from being reasonable and well-given. Indeed, the fact that near-ground concentrations remain quite low, and that high beryllium concentrations increase only in the upper atmospheric layers, probably result from the particular meteorology of the period, which probably did not favor the transport of such high concentrations to the lower tropospheric layers. In addition, the discussion of the dependence on transport is achieved only by a shift of the date of the event to a different season, without giving additional details about the particular synoptic situation characterizing those days, which leaves the interpretation of the differences between the results mostly qualitative and somehow arbitrary.

It is known that extreme SEP events in the past can be studied using 10Be isotope in polar ice cores (e.g., Usoskin et al., 2006; Mekhaldi et al., 2015; Sukhodolov et al., 2018) that typically have the (pseudo)annual resolution. Here we wanted to check, both theoretically and experimentally, whether a weaker (strong but not extreme) SEP event can be observed in high-resolution beryllium data. As far as we know this question has not been fully addressed earlier. Moreover, as our modelling shows, the effect of a SEP event on the near-ground beryllium concentrations slightly depends on the season, because of the different patterns of the large-scale dynamics. During Summer-Autumn, the low tropopause and decreased static stability of the troposphere permit a more direct coupling with the upper atmosphere opening a path for the input of the polar stratospheric beryllium to lower levels. In contrast, in Winter-Spring, the tropopause rises, and intense radiative cooling stratifies the lower troposphere closing this route.

*It is true that large-scale dynamics is strongly different in the different seasons, which on average can perhaps be described as you did above, even though there can be situations largely deviating from this average behavior. In addition, the situation that you depict is not the one that is observed*

*in reality and discussed in many scientific papers in the field. It is in fact well-known that even though the rise of the tropopause and the presence of intense convection during the warmest seasons favors the transport of upper tropospheric air to the surface, the dynamics and the instability of the coldest seasons tend to favor intense intrusions from the stratosphere to ground. However, these events of stratospheric intrusions and transport of air from the upper troposphere-lower stratosphere have to be rather intense to be observed at ground, and are usually observed only at high-altitude stations. In addition, these events do not represent the average situation, but rather a particular one extending for two or three days. Finally, even though these events are typical of the cold season, they sometimes occur even during the other season. There are lots of papers describing the dynamic of stratosphere-to-troposphere transport, a topic that though still not completely understood is not completely obscure and subject of many scientific projects/papers. Just to cite a few of the most important ones, you can see for instance: Cristofanelli et al., 2003. Stratosphere-to-troposphere transport: A model and method evaluation. Doi:10.1029/2002JD002274; Zanis et al., 2003. An estimate of the impact of stratosphere-to-troposphere transport (STT) on the lower free tropospheric ozone over the Alps using 10Be and 7Be measurements. doi:10.1029/2002JD002604; Stohl et al., 2003. Stratosphere-troposphere exchange: a review, and what we have learned from STACCATO. Doi:10.1029/2002JD002490; Tarasick et al., 2019. Quantifying stratosphere-troposphere transport of ozone using balloon-borne ozonesondes, radar windprofilers and trajectory models. Doi:10.1016/j.atmosenv.2018.10.040. My comment was aimed at clarifying exactly that if you do not present the synoptic situation characterizing the SEP event and the other seasons, you cannot be sure that the beryllium produced in the upper atmospheric layers is then transported to ground and therefore your analysis of the event during the SEP dates and in other dates in other seasons likely could be not capable of representing the effect of seasonality on beryllium concentrations.*

To conclude, the authors present lots of technical details which pertain to the methods sections (e.g., information on measurement methods for beryllium, but also modeling information) in the results. I would suggest restructuring the paper to include those details in the methods so that the results section contains only the findings of this work and their appropriate discussion.

We agree and have restructured the manuscript accordingly. Information on measurement methods for beryllium now in *Section 3*.

*Ok with this modification.*

*Specific comments*

1) Line 35: Change "from" to "form".
2) Line 40: Delete the first comma. Change "cadences" to "time resolution" or "sampling resolution" or similar. Consider revise the sentence which is quite long and not totally clear.
3) Line 45 and throughout the manuscript: Again change the term "cadence".

4) Lines 4-6 and 48-49: In general the models do not reproduce only the transport of beryllium isotopes, but also at least their deposition, and a parameterization of sources (maybe not as accurate as yours), and thus the simulated concentrations do not depend only on transport. This further means that in most cases, input meteorological fields should contain not only wind data. Please consider revise this.

5) Lines 49-50: Please check the two verbs, one is singular and the other is plural.

6) Lines 56-58: A reference is needed for this sentence.

7) Lines 58-60: This sentence, which introduces the scope of this work, is connected with the sentences at lines 107-109, 116-117 and further. I understand the need to introduce the new Section 2 for presenting the state-of-the-art models on this topic, but then wouldn't be Section 2 a part of the Introduction, and thus better presented as Section 1.1? This revision would need not only a change in the numbering, but also a better formulation of the scope of the article and its structure after presenting the gaps of past and recent works.

8) Line 82: Change "ModelE" to "model".

9) Lines 70-94: The review of earlier works is not complete. For instance, much of the work conducted by NASA with the GMI and GEOS-Chem CTMs is missing. Please consider revise conducting a careful review including a more complete overview of the modeling work to represent beryllium isotopes.

10) Line 99: The knowledge of the meteorological fields should be essential to all modeling simulations of this kind, so I do not see this as a significant gap of this methodology.

11) Line 104: Delete comma.

12) Lines 104-106: I cannot fully understand those requirements: do you mean that in general the climatological and beryllium communities do not collaborate?

13) Lines 175-176: Which data? Of which variables?

14) Line 229: Change to "with smaller absolute values".

15) Lines 291-296: References are needed.

16) Figure 11: If the caption is correct and the histograms reproduce the difference between modeled and observed 7Be values, then the title of the x-axis is not. Although the mean reported values are indicative of a low bias, the figure shows that there are cases in which the bias is high, almost equal to 1 in Chile and at Kerguelen, thus contradicting some statements by the authors. In addition, since it is known that the shape of histograms is dominated by the correct choice of the bin width, couldn't you think of a different method to represent the bias?

17) Line 348: The paper of Brattich et al. (2021) has not been added to the reference list.

18) Revise the acronym as being SPE and not SEP throughout the manuscript and figures.

---

## Referee Report (RR2)

The manuscript entitled "*Application of Chemistry-climate model SOCOL-AERv2-BEv1 to cosmogenic beryllium isotopes: Description and validation for polar regions*"
is generally well written and deals with an interesting topic, since it comes to fill a lack of knowledge regarding the behavior of 7Be and 10Be in polar regions.
There is a detailed and comprehensive analysis of data with figures and Tables to add to this complete analysis.
It is valuable for the scientific community and can be published in Geoscience_Model_Development after some minor revisions

Before my comments for some minor revisions, I'd like to make some general comments reading the revised manuscript.
I agree with the authors that "…the two beryllium isotopes are believed to have similar transport/deposition properties being different only in the production and the lifetime, thus the results of 7Be transport can be generally applied to 10Be." (lines 10-12, and 56-58 of the revised manuscript).
No further discussion is needed, no doubts, and not any further clarifications.
For sure the reviewers' comments and queries helped the authors to clarify any ambiguity.
The authors reply in details to reviewers' comments and the text is clear.

Regarding the weekly data that the authors use in their analysis, I find it absolutely scientifically correct, since daily data and discrepancies due to various reasons do not allow the scientists (the authors in our case) to proceed with reliable conclusions. It is great advantage that Finland has so many stations with so long recorded weekly data. It is almost impossible to find something similar anywhere in the world. Furthermore in lines 130-136, the authors explain more than satisfactory the reasons for using the weekly data in their analysis.

I find extremely interesting and valuable a model like the proposed one in this manuscript: *"The new model version SOCOL-AERv2-BEv1 that has been developed here for systematic modelling of 7Be and 10Be production, transport and deposition in the atmosphere and the main purpose of this work is to present a new combined model of beryllium production and transport and to confront its results with high-resolution (weekly) measurements of 7Be in near-ground air and precipitating water in polar regions has been successfully achieved."*
It is a knowledge that was missing and this work is valuable from the scientific community.
And yes, there are many papers regarding the 7Be in mid latitudes, but these data cannot be used and/or applied to 7Be at high latitudes and its behavior (similarly 10Be). And only in Polar Regions can be studied the solar energetic-particle event (SPE) and their influence in 7Be concentrations.

**Proposed revisions:**

-1-

In my opinion, there is a confusion with the names-abbreviations of the models that are mentioned in this work (***SOCOL-AERv2-BEv1, SOCOL, CCM SOCOL, SOCOL v3.0, SOCOL-AERv2, CCM SOCOL-AERv2, SOCOLv3.0:Be, SOCOLv3.0)***. See abbreviations in text at section -4- bellow.

My suggestion to the authors is to try to "keep a unique form" as much as possible with the model abbreviations, e.g. I do not understand if there is any difference between the CCM SOCOL and SOCOL. For sure if there is a difference they authors should keep the different abbreviations, but if not please keep just one.

-2-

The name of the model that is developed here **SOCOL-AERv2-BEv1**, is mentioned only once in the title and once at the line 123 of the manuscript.

**In my opinion, the name of the model must be added in the Abstract and in the Conclusions.**

-3-

Furthermore, since it is "*a new developed model*" it could be mentioned somehow in the title. But this is not mandatory and the authors will decide for the title of the final paper. It's just a proposal.

-4-

Below I mention some parts of the text with the abbreviations of the model that at least in my case produced a slight confusion.

*Title*: "…..**SOCOL-AERv2-BEv1**…."

*Abstract*
Lines 12-13: "*based on the chemistry-climate model* **SOCOL (SOlar Climate Ozone Links) v3**,…"
lines 25-26: "*Thus, a new full 3D time-dependent model, based on the* **SOCOL v3.0,** *of 7Be and 10Be atmospheric production, transport and deposition have been developed.*"

**In my opinion, the development of the new model must be mentioned in title.**

*1 Introduction*
Lines 58-60: "*Here we develop such a model to trace isotopes of 7Be and 10Be in the atmosphere based on the* **chemistry-climate model SOCOL (SOlar Climate Ozone Links) v3**, *which has been specifically modified by including modules for the production, deposition, and transport of 7Be and 10Be.*"

*2 Summary previous and existing models*
Lines 108-110: "*Here we present a new development of the* **full chemistry-climate model (CCM) SOlar Climate Ozone Links (SOCOL)** *for modelling of production, transport, and deposition of the cosmogenic isotopes of beryllium as well as its validation with the available measurements of 7Be at high-latitude locations. The* **CCM SOCOL** *is potentially capable of simulation*"

Lines 112-115: "*A recent model version* **SOCOL-AERv2**, *which simulates aerosols more realistically does not, however, include the treatment of all processes relevant to the beryllium life cycle and its applicability has not been evaluated. We have* **further upgraded the** **CCM SOCOL-AERv2** *(Feinberg et al., 2019) here, by adding the production, transport, and deposition of 7Be and 10Be isotopes from both GCR and SEPs.*"

Lines 123-124: *"The new model version **SOCOL-AERv2-BEv1** has been developed here for systematic modelling of 7Be and 10Be production, transport and deposition in the atmosphere."*

**It is the first time that the authors mention the name of the model that has been developed and presented in this work.**
**In my opinion, the name of the model must be added in the Abstract and in the Conclusions.**

*4 Model description*
Lines 171-172: *"We used an extended version of the **CCM SOCOLv3** (Stenke et al., 2013) with the aerosol module - **SOCOL-AERv2** (Feinberg et al., 2019)."*
Lines 179*: "*SOCOL uses the horizontal resolution T42, where…."*
Line 200: "… used as an input for the **SOCOL** model,…"
Line 258: "…are realistically modelled by the **CCM SOCOL** (Feinberg et al., 2019)…."

*6 Evaluation of the model by comparison with direct 7Be measurements*
Line 387: "…deposition modelled by **SOCOL**…"

*Conclusions*
Line 394: *"…The model named as **SOCOLv3.0:Be** is based on the chemistry-climate model **SOCOL**, specifically.."*
Line 411: *"Concluding, a new full 3D time-dependent model, based on **SOCOL-AERv2**, of 7Be and 10Be…."*

---

## Referee Report (RR3)

In this paper, the authors aim to assess the production rate using the CRAC: Be and the transport process of the cosmogenic Be isotopes using the Chemistry-climate model SOCOL-AERv2-BeV1. The $^7$Be concentrations in the near-ground air calculated by the model were compared with the observed concentrations at four high-latitude stations in both hemispheres from 2002 to 2008. The model calculation data in the Northern Hemisphere stations are in reasonable agreement with the observation data, especially the monthly and annual mean values.

This paper contributes to an understanding of the production and transport process of the cosmogenic Be isotopes, but some hypotheses of the gas transport process of Be isotopes are different from the common sense of transport with aerosol. Therefore, this paper is required to major revision for either modification of the transport process or explaining the substantial reason for the gaseous behavior of the cosmogenic Be isotopes.

Section 4.3

P11 L225-226: "It should be noted that we treat $^7$Be as gas only for the advective transport when the result is the same for the small particles and gas components. "- Why the $^7$Be is treated as gas not the aerosols when the result is the same. The $^7$Be is treated as the aerosols in both the previous model and the observation. Why is it not sufficient to treat the $^7$Be as an aerosol?

P12 Figure 4: Isn't the vertical axis the amount of cosmogenic Be isotopes in the atmosphere? I have never seen the concentration of the cosmogenic isotopes as $10^8$ atoms m$^{-3}$, also in the stratosphere (Jordan *et al.*, 2003).

Section 5.1

P13 L290: Since volume of the air is extremely changing with atmospheric pressure, it is recommended to use at / m$^3$ SPT as the volume unit.

P15 Figure 6: Thea color contour is unclear to represent the deposition distribution described in the text. It should be made clear that the deposition at the West of the continents is lower than other areas.

Section 5.2

P15 L316: "on the 30$^t$h day after the event "- It might be miss touch "on the 30 days after the event."

P15 L324: Miyake et al., 2018 (GRL https://doi.org/10.1029/2018GL080475) detected the SEP signal at 993-994CE of $^{10}$Be in the quasi-annual Antarctic ice core record. Is it possible that the SEP event is stronger than a few orders of magnitude?

P16 L329: Figure 9 is not atmospheric concentrations. Please check the data.

References

There is some typo in the text, and, in some cases, abbreviations are not used in the journal name. Please check again.

P22 L446: "Three-dimensional simulation of 7be in a global climate model. "- "Three-dimensional simulation of $^7$Be in a global climate model. "

P22 L465: "Modeling production and climate-related impacts on 10be concentration in ice cores. "- "Modeling production and climate-related impacts on $^{10}$Be concentration in ice cores. "

P23 L500: "Sulfur, sea salt and radionuclide aerosols in giss modele. "- "Sulfur, sea salt and radionuclide aerosols in GISS modelE."

P23 L507: "Stratosphere–troposphere exchange in a changing climate simulated with the general circulation model maecham4. "- "Stratosphere–troposphere exchange in a changing climate simulated with the general circulation model MAECHAM4. "

P23 L508: "Deposition of naturally occurring 7Be and 210Pb in Northern Finland. "- "Deposition of naturally occurring $^7$Be and $^{210}$Pb in Northern Finland. "

P24 L513:" Geomagnetic and atmospheric effects upon the cosmogenic 10be observed in polar ice. "- "Geomagnetic and atmospheric effects upon the cosmogenic $^{10}$Be observed in polar ice. "

P24 L522: "Extended versions of the convective parametrization scheme at ecmwf and their impact on the mean and transient activity of the model in the tropics"- "Extended versions of the convective parametrization scheme at ECMWF and their impact on the mean and transient activity of the model in the tropics"

P24 L525: "Geophysical Research Letters" is not an abbreviation.

P35 L558: "Global cloud and precipitation chemistry and wet deposition: tropospheric model simulations with echam5/messy1. Atmospheric Chemistry and Physics "- "Global cloud and precipitation chemistry and wet deposition: tropospheric model simulations with EHCAM5/MESSy1. ". "Atmospheric Chemistry and Physics" is not an abbreviation.

L26 L597: "atmospheric 7Be in Europe"- "atmospheric $^7$Be in Europe"

---

## Author Response (AR2)

**Reviewer 1**

We thank the Reviewer for her/his detailed comment. We address all her comments below, where highlighted text is the Reviewer's comment followed by the unformatted text of our reply.

*In this case I am rather satisfied with the revision. There is still confusion regarding the application of the model to beryllium isotopes and then its presentation and validation considering just Be-7.*
*Also, I do not completely understand the connection between a beryllium model with eventual application to Be-10 data with the choice of high-latitude regions and annual timescales. Could you please better specify this?*

As explained in the revised introduction, our eventual goal is to understand the production/transport of 10Be measured in polar ice cores with annual resolution. 10Be is a primary isotope to study cosmic-ray and SEP variability on the long-term scale before the era of direct measurements (e.g., Beer et al., 2012; Usoskin, 2017). Unfortunately, the existing models are unable to fully reproduce the measured data in polar ice, and higher accuracy of models is needed. This is why we are primarily interested in polar regions and annual time scales. The purpose of this paper is to check and validate the ability of the model to perform the first step, viz. production and transport of beryllium. However, the model output can hardly be compared directly to data because 10 Be is not measured in the air but rather as 10Be concentration in the ice where it is additionally affected by the deposition and possible post-depositional effects. These effects are left for forthcoming studies and not touched here. The only way to validate a beryllium transport model is to use 7Be which is routinely measured in air samples. 7Be is similar to 10Be in production and transport but has a different lifetime which is easy to account for. Thus, we compared the model output for 7Be with the measured near-ground air concentrations focussing upon high-latitude regions and annual timescale, as explained above. We are happy that Reviewers 2 and 3 understand our motivation. As Reviewer 2 states "And yes, there are many papers regarding the 7Be in mid latitudes, but these data cannot be used and/or applied to 7Be at high latitudes and its behavior (similarly 10Be). And only in Polar Regions can be studied the solar energetic-particle event (SPE) and their influence in 7Be concentrations."

*As commented below in the specific comments, there are still some deficiencies in the added Section 2. Indeed, the review of earlier models is far from being complete, missing important works on this subject. It is true that this field is still widely open for improvements, but the state-of-the-art in this research field has to be consistently described. Secondly, to me the newly added Section is part of the Introduction section, being fundamental for the reader to understand the need of this particular work and its scope. Therefore, to be the introduction of this as Section 2 generates confusion and does not fully help to address the above reported issues.*

We disagree with this comment. Moving discussion of previous models to the Introduction would make it heavily distracting the reader's attention from the formulation of the aims of the paper. Instead, a separate focused Section 2 makes it easier to read.

*As commented below, the histograms shown in Figure 11 actually indicate that apart from the mean value of the bias, there are many cases, especially in southern stations, when the bias is actually higher and close to 1. Indeed, the mean low value could result from the distribution of the bias. In addition, the histogram may not be the best way to represent the bias, since the shape of the histogram is dominated by the choice of the bin width. As I previously mentioned, there are a set of statistical parameters to calculate and validate a model. In addition, it would be interesting to evaluate the pattern of the bias to investigate if it is higher during particular seasons or if conversely the high biases are randomly distributed.*

We believe that this comment is caused by confusion. Figure 11 shows that there is no significant bias between the model results and measurements. The Reviewer apparently confuses the *bias* which is a systematic displacement and random *deviations*. The histograms imply that there are no biases but deviations

can be large. However, these discrepancies are not worrisome for our purpose because they occur on a synoptic timescale which is known to be not reproduced by the general circulation models (e.g., Usoskin et al., 2009, Brattich et al., 2017). Deviations are larger during the local summer season, in agreement with other models (Brattich et al., 2017). As one can see from Fig.10, the agreement at the annual scale is always good. The exact shape of the histogram may indeed depend on the bin width, but the metrics of the distribution (the mean and standard deviation) are defined unambiguously by the dataset and do NOT depend on the parameters of the histogram.

In our view, the presently used standard set of statistical metrics is sufficient to validate our model: the correlation/coherence analysis confirms that the modelled and measured variabilities of 7Be concentrations agree very well on the annual scale, while the distributions of the residuals confirm the absence of any significant bias between the model and the data.

*Ok with this addition. However, it is still not clear which kind of meteorological data have been used.*

CCM SOCOL uses ECHAM5 nudged with ERA-Interim (eraiaT42L39) reanalyses. Full nudging is a linear relaxation of thermodynamic parameters: temperature, divergence and vorticity of the wind field.

*It is true that large-scale dynamics is strongly different in the different seasons, which on average can perhaps be described as you did above, even though there can be situations largely deviating from this average behaviour. In addition, the situation that you depict is not the one that is observed in reality and discussed in many scientific papers in the field. It is in fact well-known that even though the rise of the tropopause and the presence of intense convection during the warmest seasons favor the transport of upper tropospheric air to the surface, the dynamics and the instability of the coldest seasons tend to favor intense intrusions from the stratosphere to ground. However, these events of stratospheric intrusions and transport of air from the upper troposphere-lower stratosphere have to be rather intense to be observed at ground, and are usually observed only at high-altitude stations. In addition, these events do not represent the average situation, but rather a particular one extending for two or three days. Finally, even though these events are typical of the cold season, they sometimes occur even during the other season. There are lots of papers describing the dynamic of stratosphere-to-troposphere transport, a topic that though still not completely understood is not completely obscure and subject of many scientific projects/papers. Just to cite a few of the most important ones, you can see for instance: Cristofanelli et al., 2003. Stratosphere-to-troposphere transport: A model and method evaluation. Doi:10.1029/2002JD002274; Zanis et al., 2003. An estimate of the impact of stratosphere-to-troposphere transport (STT) on the lower free tropospheric ozone over the Alps using 10Be and 7Be measurements. doi:10.1029/2002JD002604; Stohl et al., 2003. Stratosphere-troposphere exchange: a review, and what we have learned from STACCATO. Doi:10.1029/2002JD002490; Tarasick et al., 2019. Quantifying stratosphere-troposphere transport of ozone using balloon-borne ozonesondes, radar windprofilers and trajectory models. Doi:10.1016/j.atmosenv.2018.10.040.*

*My comment was aimed at clarifying exactly that if you do not present the synoptic situation characterizing the SEP event and the other seasons, you cannot be sure that the beryllium produced in the upper atmospheric layers is then transported to ground and therefore your analysis of the event during the SEP dates and in other dates in other seasons likely could be not capable of representing the effect of seasonality on beryllium concentrations.*

An analysis of the synoptic situations would be definitely of interest, but it lies beyond the scope of this manuscript, which aims at the description and evaluation of the new model, as explained above, with focus upon high-latitude regions and annual timescale. For our case, the representation of more robust systematic seasonal STE changes is more important.

**Specific comments**

1) Line 35: Change "from" to "form".

Done.

2) Line 40: Delete the first comma. Change "cadences" to "time resolution" or "sampling resolution" or similar. Consider revise the sentence which is quite long and not totally clear.

It is changed to "time resolution"

3) Line 45 and throughout the manuscript: Again change the term "cadence".

It is changed to "time resolution".

4) Lines 4-6 and 48-49: In general the models do not reproduce only the transport of beryllium isotopes, but also at least their deposition, and a parameterization of sources (maybe not as accurate as yours), and thus the simulated concentrations do not depend only on transport. This further means that in most cases, input meteorological fields should contain not only wind data. Please consider revise this.

Lines 4-6: While transport of $^7$Be can be modelled with high accuracy using the known meteorological (wind) fields, atmospheric transport of $^{10}$Be was typically modelled using case-study specific simulations or simplified box models based on parametrizations.

We rephrased it as

Lines 5-7: While transport and deposition of $^7$Be can be modelled with high accuracy using the known meteorological fields, $^{10}$Be was typically modelled using case-study specific simulations or simplified box models based on parametrizations.

Line 49: Its atmospheric transport can be modelled with high accuracy using the known meteorological (wind) fields.

We rephrased it as

Line 49-50: Its atmospheric transport can be modelled with high accuracy using the known meteorological fields.

5) Lines 49-50: Please check the two verbs, one is singular and the other is plural.

Done, see point 4.

6) Lines 56-58: A reference is needed for this sentence.

The following reference has been added to the list:

U. Heikkilä, J. Beer, J. Feichter. Modeling cosmogenic radionuclides 10Be and 7Be during the Maunder Minimum using the ECHAM5-HAM General Circulation Model. Atmospheric Chemistry and Physics, European Geosciences Union, 2008, 8 (10), pp.2797-2809. ffhal-00296559

7) Lines 58-60: This sentence, which introduces the scope of this work, is connected with the sentences at lines 107-109, 116-117 and further. I understand the need to introduce the new Section 2 for presenting the state-of-the-art models on this topic, but then wouldn't be Section 2 a part of the Introduction, and thus better presented as Section 1.1? This revision would need not only a change in the numbering, but also a better formulation of the scope of the article and its structure after presenting the gaps of past and recent works.

We do not agree with this suggestion (see above).

8) Line 82: Change "ModelE" to "model".

The term "ModelE" is correct, see description at https://www.giss.nasa.gov/tools/modelE/

9) Lines 70-94: The review of earlier works is not complete. For instance, much of the work conducted by NASA with the GMI and GEOS-Chem CTMs is missing. Please consider revise conducting a careful review including a more complete overview of the modeling work to represent beryllium isotopes.

We have added information about GMI CTM in the review of earlier works. However, the CTM approach applies only to years where the meteorological fields are known, while CCM can work self-consistently without prescribed meteorology. Accordingly, CTM models cannot be used to study beryllium isotopes (mainly, $^{10}$Be) in the past or future scenarios, that is the main aim of this work.

10) Line 99: The knowledge of the meteorological fields should be essential to all modeling simulations of this kind, so I do not see this as a significant gap of this methodology.

We are talking about prescribed observation-based meteorology, which does not exist for the past and future, while our primary goal is to model $^{10}$Be production/transport/deposition for the past data and future-projection scenarios.

We rephrased line 127 as:

The latter approach is applicable only to recent years where the observed meteorology is known.

Moreover, this approach does not include either stratospheric dynamics or depositional processes and, therefore, is not suitable for $^{10}$Be in polar ice.

11) Line 104: Delete comma.

Done.

12) Lines 104-106: I cannot fully understand those requirements: do you mean that in general, the climatological and beryllium communities do not collaborate?

Unfortunately, yes. The collaboration ought to be much closer.

13) Lines 175-176: Which data? Of which variables?

Full nudging means a linear relaxation of thermodynamic parameters: temperature, divergence and vorticity of the wind field.

14) Line 229: Change to "with smaller absolute values".

Done.

15) Lines 291-296: References are needed.

The following reference has been added:

Terzi, L., Wotawa, G., Schoeppner, M. et al. Radioisotopes demonstrate changes in global atmospheric circulation possibly caused by global warming. Sci Rep 10, 10695 (2020). https://doi.org/10.1038/s41598-020-66541-5.

16) Figure 11: If the caption is correct and the histograms reproduce the difference between modeled and observed 7Be values, then the title of the x-axis is not. Although the mean reported values are indicative of a low bias, the figure shows that there are cases in which the bias is high, almost equal to 1 in Chile and at Kerguelen, thus contradicting some statements by the authors. In addition, since it is known that the shape of histograms is dominated by the correct choice of the bin width, couldn't you think of a different method to represent the bias?

We don't understand why the Reviewer believes that the title of the x-axis is not correct. The Figure caption clearly says that this is the difference between modelled and measured 7Be weekly activities, and the unit of

the difference is indeed the activity, viz. mBq/m3, as written in the X-axis title. As discussed above, the Reviewer apparently confuses concepts of "bias" (systematic discrepancy) and "deviation" (instant difference). This Figure shows that there is no significant/notable bias between the model results and the data, but the deviations can be indeed large on the synoptic (weekly) timescale. However, since we are mainly interested in the annual timescale, this is not a problem. While the shape of the histogram may slightly depend on the bin width, the parameters of the distribution of the residuals (the mean and the standard deviation) are unambiguously defined by the data and have nothing to do with the histogram bin width.

17) Line 348: The paper of Brattich et al. (2021) has not been added to the reference list.

The paper of Brattich et al. (2021) was added.

18) Revise the acronym as being SPE and not SEP throughout the manuscript and figures.
It is fixed as SPE throughout the paper.

**Reviewer 2**

We are thankful to this Reviewer for her thorough and in-depth comments, deep knowledge of the field and positive attitude.

**Proposed revisions:**
-1-
In my opinion, there is a confusion with the names-abbreviations of the models that are mentioned in this work (**SOCOL-AERv2-BEv1, SOCOL, CCM SOCOL, SOCOL v3.0, SOCOL-AERv2, CCM SOCOLAERv2, SOCOLv3.0:Be, SOCOLv3.0)**. See abbreviations in text at section -4- bellow.
My suggestion to the authors is to try to "keep a unique form" as much as possible with the model abbreviations, e.g. I do not understand if there is any difference between the CCM SOCOL and SOCOL. For sure if there is a difference they authors should keep the different abbreviations, but if not please keep just one.
-2-
The name of the model that is developed here **SOCOL-AERv2-BEv1**, is mentioned only once in the title and once at the line 123 of the manuscript.
**In my opinion, the name of the model must be added in the Abstract and in the Conclusions.**
-3-
Furthermore, since it is "*a new developed model*" it could be mentioned somehow in the title. But this is not mandatory and the authors will decide for the title of the final paper. It's just a proposal.
-4-
Below I mention some parts of the text with the abbreviations of the model that at least in my case produced a slight confusion.
*Title*: "……**SOCOL-AERv2-BEv1**…."
*Abstract*
Lines 12-13: "*based on the chemistry-climate model* **SOCOL (SOlar Climate Ozone Links) v3**,…"
lines 25-26: "*Thus, a new full 3D time-dependent model, based on the* **SOCOL v3.0,** *of 7Be and 10Be atmospheric production, transport and deposition have been developed.*"
**In my opinion, the development of the new model must be mentioned in title.**
*1 Introduction*
Lines 58-60: "*Here we develop such a model to trace isotopes of 7Be and 10Be in the atmosphere*

based on the **chemistry-climate model SOCOL (SOlar Climate Ozone Links) v3**, *which has been specifically modified by including modules for the production, deposition, and transport of 7Be and 10Be."*

*2 Summary previous and existing models*

Lines 108-110: *"Here we present a new development of the **full chemistry-climate model (CCM) SOlar Climate Ozone Links (SOCOL)** for modelling of production, transport, and deposition of the cosmogenic isotopes of beryllium as well as its validation with the available measurements of 7Be at high-latitude locations. The **CCM SOCOL** is potentially capable of simulation"*

Lines 112-115: *"A recent model version **SOCOL-AERv2**, which simulates aerosols more realistically does not, however, include the treatment of all processes relevant to the beryllium life cycle and its applicability has not been evaluated. We have **further upgraded the CCM SOCOL-AERv2** (Feinberg et al., 2019) here, by adding the production, transport, and deposition of 7Be and 10Be isotopes from both GCR and SEPs."*

Lines 123-124: *"The new model version **SOCOL-AERv2-BEv1** has been developed here for systematic modelling of 7Be and 10Be production, transport and deposition in the atmosphere."*

**It is the first time that the authors mention the name of the model that has been developed and presented in this work.**

**In my opinion, the name of the model must be added in the Abstract and in the Conclusions.**

*4 Model description*

Lines 171-172: *"We used an extended version of the **CCM SOCOLv3** (Stenke et al., 2013) with the aerosol module - **SOCOL-AERv2** (Feinberg et al., 2019)."*

Lines 179: *"SOCOL uses the horizontal resolution T42, where…."*

Line 200: "… used as an input for the **SOCOL** model,…"

Line 258: "…are realistically modelled by the **CCM SOCOL** (Feinberg et al., 2019)…."

*6 Evaluation of the model by comparison with direct 7Be measurements*

Line 387: "…deposition modelled by **SOCOL**…"

*Conclusions*

Line 394: *"…The model named as **SOCOLv3.0:Be** is based on the chemistry-climate model **SOCOL**, specifically.."*

Line 411: *"Concluding, a new full 3D time-dependent model, based on **SOCOL-AERv2**, of 7Be and 10Be…."*

We thank this Reviewer for her valuable suggestion. We now use only two terms "CCM SOCOL" for the basic version of the model, and CCM "SOCOL-AERv2-BE" for the specific modified version.

**Reviewer 3**

We thank the Reviewer for his/her useful comments and deep knowledge. We address all the specific comments below.

P11 L225-226: "It should be noted that we treat 7Be as gas only for the advective transport when the result is the same for the small particles and gas components. "- Why the 7Be is treated as gas not the aerosols when the result is the same. The 7Be is treated as the aerosols in both the previous model and the observation. Why is it not sufficient to treat the 7Be as an aerosol?

We agree that the previous formulation was confusing and potentially misleading. In the model, gases and small aerosol particles are transported as passive tracers. Thus, although beryllium is indeed attached to aerosols and considered so in the model, its transport is modelled in a way similar to gases, only for the advective transport. In order to avoid further confusion, we have re-written the text in several places:

(line 271)

It should be noted that we treat $^7$Be as a passive tracer only for the advective transport when the result is the same for the small particles and gas components.

Line (274)

In this study, the advective transport of 7Be and 10Be as passive tracers was performed using Flux-Form Semi-Lagrangian Transport Schemes (Lin and Rood, 1996) embedded in ECHAM5.

Line (288)

It means that beryllium atoms are considered as passive tracer only for the transport process, while with respect to the dry and wet depositions, they are treated as sulfates.

Line (296)

Scavenging coefficients for all tracers are calculated based on Henry's law equilibrium constants.

P12 Figure 4: Isn't the vertical axis the amount of cosmogenic Be isotopes in the atmosphere? I have never seen the concentration of the cosmogenic isotopes as 108 atoms m-3, also in the stratosphere (Jordan et al., 2003).

We agree that the plot and the units were confusing. It has been redone and now shows the globally averaged columnar content of beryllium as the number of atoms per cm2 of the Earth's surface.

P13 L290: Since volume of the air is extremely changing with atmospheric pressure, it is recommended to use at / m3 SPT as the volume unit.

We have reploted fig. 4 using atoms per cm$^2$ of the Earth's surface.

P15 Figure 6: Thea color contour is unclear to represent the deposition distribution described in the text. It should be made clear that the deposition at the West of the continents is lower than other areas.

It has been added in lines 317-319.

P15 L316: "on the 30th day after the event "- It might be miss touch "on the 30 days after the event."

Done.

P15 L324: Miyake et al., 2018 (GRL https://doi.org/10.1029/2018GL080475) detected the SEP signal at 993-994CE of 10Be in the quasi-annual Antarctic ice core record. Is it possible that the SEP event I stronger than a few orders of magnitude?

Indeed, several events have been detected in the past, particularly in annual 10Be ice-core data. The strongest event of 774-775 CE was a factor ~300 greater than the event of 20-Jan-2005 considered here. We modelled the production and transport of beryllium for a major solar energetic-particle event of 20-

Jan-2005, which may serve as a reference event for historically known extreme SPEs. We have updated the text accordingly.

P16 L329: Figure 9 is not atmospheric concentrations. Please check the data.

We have checked our plot and found it consistent with similar results shown, e.g., in Heikkilä, U. and Smith, A. M.: Influence of model resolution on the atmospheric transport of 10Be, Atmos. Chem. Phys., 12, 10601–10612, https://doi.org/10.5194/acp-12-10601-2012, 2012 (see fig. 8 for L39 therein). The latter is provided only for GCR, while our model includes also SPE, thus, the values in the upper atmosphere are high.

P22 L446: "Three-dimensional simulation of 7be in a global climate model. "- "Three-dimensional simulation of 7Be in a global climate model. "

Done.

P22 L465: "Modeling production and climate-related impacts on 10be concentration in ice cores. "- "Modeling production and climate-related impacts on 10Be concentration in ice cores. "

Done.

P23 L500: "Sulfur, sea salt and radionuclide aerosols in giss modele. "- "Sulfur, sea salt and radionuclide aerosols in GISS modelE."

Done.

P23 L507: "Stratosphere–troposphere exchange in a changing climate simulated with the general circulation model maecham4. "- "Stratosphere–troposphere exchange in a changing climate simulated with the general circulation model MAECHAM4. "'

Done

P23 L508: "Deposition of naturally occurring 7Be and 210Pb in Northern Finland. "- "Deposition of naturally occurring 7Be and 210Pb in Northern Finland. "

Done

P24 L513:" Geomagnetic and atmospheric effects upon the cosmogenic 10be observed in polar ice. "- "Geomagnetic and atmospheric effects upon the cosmogenic 10Be observed in polar ice. "

Done.

P24 L522: "Extended versions of the convective parametrization scheme at ecmwf and their impact on the mean and transient activity of the model in the tropics"- "Extended versions of the convective parametrization scheme at ECMWF and their impact on the mean and transient activity of the model in the tropics"

Done.

P24 L525: "Geophysical Research Letters" is not an abbreviation.

Done.

P35 L558: "Global cloud and precipitation chemistry and wet deposition: tropospheric model simulations with echam5/messy1. Atmospheric Chemistry and Physics "- "Global cloud and precipitation chemistry and wet deposition: tropospheric model simulations with EHCAM5/MESSy1. ". "Atmospheric Chemistry and Physics" is not an abbreviation.

Done.

L26 L597: "atmospheric 7Be in Europe"- "atmospheric 7Be in Europe"

Done